# GRADIENT-BASED NEURAL DAG LEARNING

**Sébastien Lachapelle, Philippe Brouillard, Tristan Deleu & Simon Lacoste-Julien**[†]

Mila & DIRO
Université de Montréal

## ABSTRACT

We propose a novel score-based approach to learning a directed acyclic graph (DAG) from observational data. We adapt a recently proposed continuous constrained optimization formulation to allow for nonlinear relationships between variables using neural networks. This extension allows to model complex interactions while avoiding the combinatorial nature of the problem. In addition to comparing our method to existing continuous optimization methods, we provide missing empirical comparisons to nonlinear greedy search methods. On both synthetic and real-world data sets, this new method outperforms current continuous methods on most tasks, while being competitive with existing greedy search methods on important metrics for causal inference.

## 1    INTRODUCTION

Structure learning and causal inference have many important applications in different areas of science such as genetics (Koller & Friedman, 2009; Peters et al., 2017), biology (Sachs et al., 2005) and economics (Pearl, 2009). *Bayesian networks* (BN), which encode conditional independencies using *directed acyclic graphs* (DAG), are powerful models which are both interpretable and computationally tractable. *Causal graphical models* (CGM) (Peters et al., 2017) are BNs which support *interventional* queries like: *What will happen if someone external to the system intervenes on variable X?* Recent work suggests that causality could partially solve challenges faced by current machine learning systems such as robustness to out-of-distribution samples, adaptability and explainability (Pearl, 2019; Magliacane et al., 2018). However, structure and causal learning are daunting tasks due to both the combinatorial nature of the space of structures (the number of DAGs grows *super exponentially* with the number of nodes) and the question of *structure identifiability* (see Section 2.2). Nevertheless, these graphical models known qualities and promises of improvement for machine intelligence renders the quest for structure/causal learning appealing.

The typical motivation for learning a causal graphical model is to predict the effect of various interventions. A CGM can be best estimated when given interventional data, but interventions are often costly or impossible to obtained. As an alternative, one can use exclusively observational data and rely on different assumptions which make the graph identifiable from the distribution (see Section 2.2). This is the approach employed in this paper.

We propose a score-based method (Koller & Friedman, 2009) for structure learning named GraN-DAG which makes use of a recent reformulation of the original *combinatorial problem* of finding an optimal DAG into a *continuous constrained optimization problem*. In the original method named NOTEARS (Zheng et al., 2018), the directed graph is encoded as a *weighted adjacency matrix* which represents coefficients in a linear *structural equation model* (SEM) (Pearl, 2009) (see Section 2.3) and enforces acyclicity using a constraint which is both efficiently computable and easily differentiable, thus allowing the use of numerical solvers. This continuous approach improved upon popular methods while avoiding the design of greedy algorithms based on heuristics.

Our first contribution is to extend the framework of Zheng et al. (2018) to deal with nonlinear relationships between variables using neural networks (NN) (Goodfellow et al., 2016). To adapt the acyclicity constraint to our nonlinear model, we use an argument similar to what is used in

---

[†]Canada CIFAR AI Chair
Correspondence to: `sebastien.lachapelle@umontreal.ca`

Zheng et al. (2018) and apply it first at the level of *neural network paths* and then at the level of *graph paths*. Although GraN-DAG is general enough to deal with a large variety of parametric families of conditional probability distributions, our experiments focus on the special case of nonlinear Gaussian *additive noise models* since, under specific assumptions, it provides appealing theoretical guarantees easing the comparison to other graph search procedures (see Section 2.2 & 3.3). On both synthetic and real-world tasks, we show GraN-DAG often outperforms other approaches which leverage the continuous paradigm, including DAG-GNN (Yu et al., 2019), a recent nonlinear extension of Zheng et al. (2018) which uses an evidence lower bound as score.

Our second contribution is to provide a missing empirical comparison to existing methods that support nonlinear relationships but tackle the optimization problem in its discrete form using greedy search procedures, namely CAM (Bühlmann et al., 2014) and GSF (Huang et al., 2018). We show that GraN-DAG is competitive on the wide range of tasks we considered, while using pre- and post-processing steps similar to CAM.

We provide an implementation of GraN-DAG here.

## 2 BACKGROUND

Before presenting GraN-DAG, we review concepts relevant to structure and causal learning.

### 2.1 CAUSAL GRAPHICAL MODELS

We suppose the natural phenomenon of interest can be described by a random vector $X \in \mathbb{R}^d$ entailed by an underlying CGM $(P_X, \mathcal{G})$ where $P_X$ is a probability distribution over $X$ and $\mathcal{G} = (V, E)$ is a DAG (Peters et al., 2017). Each node $j \in V$ corresponds to exactly one variable in the system. Let $\pi_j^{\mathcal{G}}$ denote the set of parents of node $j$ in $\mathcal{G}$ and let $X_{\pi_j^{\mathcal{G}}}$ denote the random vector containing the variables corresponding to the parents of $j$ in $\mathcal{G}$. Throughout the paper, we assume there are no hidden variables. In a CGM, the distribution $P_X$ is said to be *Markov* to $\mathcal{G}$, i.e. we can write the probability density function (pdf) of $P_X$ as $p(x) = \prod_{j=1}^d p_j(x_j|x_{\pi_j^{\mathcal{G}}})$ where $p_j(x_j|x_{\pi_j^{\mathcal{G}}})$ is the conditional pdf of variable $X_j$ given $X_{\pi_j^{\mathcal{G}}}$. A CGM can be thought of as a BN in which directed edges are given a causal meaning, allowing it to answer queries regarding *interventional distributions* (Koller & Friedman, 2009).

### 2.2 STRUCTURE IDENTIFIABILITY

In general, it is impossible to recover $\mathcal{G}$ given only samples from $P_X$, i.e. without *interventional data*. It is, however, customary to rely on a set of assumptions to render the structure fully or partially *identifiable*.

**Definition 1** *Given a set of assumptions $A$ on a CGM $\mathcal{M} = (P_X, \mathcal{G})$, its graph $\mathcal{G}$ is said to be identifiable from $P_X$ if there exists no other CGM $\tilde{\mathcal{M}} = (\tilde{P}_X, \tilde{\mathcal{G}})$ satisfying all assumptions in $A$ such that $\tilde{\mathcal{G}} \neq \mathcal{G}$ and $\tilde{P}_X = P_X$.*

There are many examples of graph identifiability results for continuous variables (Peters et al., 2014; Peters & Bühlman, 2014; Shimizu et al., 2006; Zhang & Hyvärinen, 2009) as well as for discrete variables (Peters et al., 2011). These results are obtained by assuming that the conditional densities belong to a specific parametric family. For example, if one assumes that the distribution $P_X$ is entailed by a structural equation model of the form

$$X_j := f_j(X_{\pi_j^{\mathcal{G}}}) + N_j \quad \text{with } N_j \sim \mathcal{N}(0, \sigma_j^2) \ \forall j \in V \tag{1}$$

where $f_j$ is a nonlinear function satisfying some mild regularity conditions and the noises $N_j$ are mutually independent, then $\mathcal{G}$ is identifiable from $P_X$ (see Peters et al. (2014) for the complete theorem and its proof). This is a particular instance of *additive noise models* (ANM). We will make use of this result in our experiments in Section 4.

One can consider weaker assumptions such as *faithfulness* (Peters et al., 2017). This assumption allows one to identify, not $\mathcal{G}$ itself, but the *Markov equivalence class* to which it belongs (Spirtes

et al., 2000). A Markov equivalence class is a set of DAGs which encode exactly the same set of conditional independence statements and can be characterized by a graphical object named a *completed partially directed acyclic graph* (CPDAG) (Koller & Friedman, 2009; Peters et al., 2017). Some algorithms we use as baselines in Section 4 output only a CPDAG.

## 2.3 NOTEARS: CONTINUOUS OPTIMIZATION FOR STRUCTURE LEARNING

Structure learning is the problem of learning $\mathcal{G}$ using a data set of $n$ samples $\{x^{(1)}, ..., x^{(n)}\}$ from $P_X$. Score-based approaches cast this problem as an optimization problem, i.e. $\hat{\mathcal{G}} = \arg\max_{\mathcal{G} \in \text{DAG}} \mathcal{S}(\mathcal{G})$ where $\mathcal{S}(\mathcal{G})$ is a regularized maximum likelihood under graph $\mathcal{G}$. Since the number of DAGs is super exponential in the number of nodes, most methods rely on various heuristic greedy search procedures to approximately solve the problem (see Section 5 for a review). We now present the work of Zheng et al. (2018) which proposes to cast this combinatorial optimization problem into a continuous constrained one.

To do so, the authors propose to encode the graph $\mathcal{G}$ on $d$ nodes as a weighted adjacency matrix $U = [u_1| \dots |u_d] \in \mathbb{R}^{d \times d}$ which represents (possibly negative) coefficients in a linear SEM of the form $X_j := u_j^\top X + N_i \ \ \forall j$ where $N_j$ is a noise variable. Let $\mathcal{G}_U$ be the directed graph associated with the SEM and let $A_U$ be the (binary) adjacency matrix associated with $\mathcal{G}_U$. One can see that the following equivalence holds:

$$(A_U)_{ij} = 0 \iff U_{ij} = 0 \tag{2}$$

To make sure $\mathcal{G}_U$ is acyclic, the authors propose the following constraint on $U$:

$$\operatorname{Tr} e^{U \odot U} - d = 0 \tag{3}$$

where $e^M \triangleq \sum_{k=0}^{\infty} \frac{M^k}{k!}$ is the *matrix exponential* and $\odot$ is the Hadamard product.

To see why this constraint characterizes acyclicity, first note that $(A_U{}^k)_{jj}$ is the number of cycles of length $k$ passing through node $j$ in graph $\mathcal{G}_U$. Clearly, for $\mathcal{G}_U$ to be acyclic, we must have $\operatorname{Tr} A_U{}^k = 0$ for $k = 1, 2, ..., \infty$. By equivalence (2), this is true when $\operatorname{Tr}(U \odot U)^k = 0$ for $k = 1, 2, ..., \infty$. From there, one can simply apply the definition of the matrix exponential to see why constraint (3) characterizes acyclicity (see Zheng et al. (2018) for the full development).

The authors propose to use a regularized negative least square score (maximum likelihood for a Gaussian noise model). The resulting continuous constrained problem is

$$\max_U \mathcal{S}(U, \mathbf{X}) \triangleq -\frac{1}{2n}\|\mathbf{X} - \mathbf{X}U\|_F^2 - \lambda\|U\|_1 \quad \text{s.t.} \quad \operatorname{Tr} e^{U \odot U} - d = 0 \tag{4}$$

where $\mathbf{X} \in \mathbb{R}^{n \times d}$ is the design matrix containing all $n$ samples. The nature of the problem has been drastically changed: we went from a combinatorial to a continuous problem. The difficulties of combinatorial optimization have been replaced by those of non-convex optimization, since the feasible set is non-convex. Nevertheless, a standard numerical solver for constrained optimization such has an *augmented Lagrangian method* (Bertsekas, 1999) can be applied to get an approximate solution, hence there is no need to design a greedy search procedure. Moreover, this approach is more global than greedy methods in the sense that the whole matrix $U$ is updated at each iteration. Continuous approaches to combinatorial optimization have sometimes demonstrated improved performance over discrete approaches in the literature (see for example Alayrac et al. (2018, §5.2) where they solve the multiple sequence alignment problem with a continuous optimization method).

## 3 GRAN-DAG: GRADIENT-BASED NEURAL DAG LEARNING

We propose a new nonlinear extension to the framework presented in Section 2.3. For each variable $X_j$, we learn a fully connected neural network with $L$ hidden layers parametrized by $\phi_{(j)} \triangleq \{W_{(j)}^{(1)}, \dots, W_{(j)}^{(L+1)}\}$ where $W_{(j)}^{(\ell)}$ is the $\ell$th weight matrix of the $j$th NN (biases are omitted for clarity). Each NN takes as input $X_{-j} \in \mathbb{R}^d$, i.e. the vector $X$ with the $j$th component masked to zero, and outputs $\theta_{(j)} \in \mathbb{R}^m$, the $m$-dimensional parameter vector of the desired distribution family

for variable $X_j$.[1] The fully connected NNs have the following form

$$\theta_{(j)} \triangleq W_{(j)}^{(L+1)} g(\dots g(W_{(j)}^{(2)} g(W_{(j)}^{(1)} X_{-j})) \dots) \;\; \forall j \tag{5}$$

where $g$ is a nonlinearity applied element-wise. Note that the evaluation of all NNs can be parallelized on GPU. Distribution families need not be the same for each variable. Let $\phi \triangleq \{\phi_{(1)}, \dots, \phi_{(d)}\}$ represents all parameters of all $d$ NNs. Without any constraint on its parameter $\phi_{(j)}$, neural network $j$ models the conditional pdf $p_j(x_j | x_{-j}; \phi_{(j)})$. Note that the product $\prod_{j=1}^d p_j(x_j | x_{-j}; \phi_{(j)})$ does not integrate to one (i.e. it is not a joint pdf), since it does not decompose according to a DAG. We now show how one can constrain $\phi$ to make sure the product of all conditionals outputted by the NNs is a joint pdf. The idea is to define a new weighted adjacency matrix $A_\phi$ similar to the one encountered in Section 2.3, which can be directly used inside the constraint of Equation 3 to enforce acyclicity.

## 3.1  Neural network connectivity

Before defining the weighted adjacency matrix $A_\phi$, we need to focus on how one can make some NN outputs unaffected by some inputs. Since we will discuss properties of a single NN, we drop the NN subscript $(j)$ to improve readability.

We will use the term *neural network path* to refer to a computation path in a NN. For example, in a NN with two hidden layers, the sequence of weights $(W_{h_1 i}^{(1)}, W_{h_2 h_1}^{(2)}, W_{k h_2}^{(3)})$ is a NN path from input $i$ to output $k$. We say that a NN path is *inactive* if at least one weight along the path is zero. We can loosely interpret the *path product* $|W_{h_1 i}^{(1)}| |W_{h_2 h_1}^{(2)}| |W_{k h_2}^{(3)}| \geq 0$ as the strength of the NN path, where a path product is equal to zero if and only if the path is inactive. Note that if all NN paths from input $i$ to output $k$ are inactive (i.e. the sum of their path products is zero), then output $k$ does not depend on input $i$ anymore since the information in input $i$ will never reach output $k$. The sum of all path products from input $i$ to output $k$ for all input $i$ and output $k$ can be easily computed by taking the following matrix product.

$$C \triangleq |W^{(L+1)}| \dots |W^{(2)}| |W^{(1)}| \in \mathbb{R}_{\geq 0}^{m \times d} \tag{6}$$

where $|W|$ is the element-wise absolute value of $W$. Let us name $C$ the *neural network connectivity matrix*. It can be verified that $C_{ki}$ is the sum of all NN path products from input $i$ to output $k$. This means it is sufficient to have $C_{ki} = 0$ to render output $k$ independent of input $i$.

Remember that each NN in our model outputs a parameter vector $\theta$ for a conditional distribution and that we want the product of all conditionals to be a valid joint pdf, i.e. we want its corresponding directed graph to be acyclic. With this in mind, we see that it could be useful to make a certain parameter $\theta$ not dependent on certain inputs of the NN. To have $\theta$ independent of variable $X_i$, it is sufficient to have $\sum_{k=1}^m C_{ki} = 0$.

## 3.2  A weighted adjacency matrix

We now define a weighted adjacency matrix $A_\phi$ that can be used in constraint of Equation 3.

$$(A_\phi)_{ij} \triangleq \begin{cases} \sum_{k=1}^m \left(C_{(j)}\right)_{ki}, & \text{if } j \neq i \\ 0, & \text{otherwise} \end{cases} \tag{7}$$

where $C_{(j)}$ denotes the connectivity matrix of the NN associated with variable $X_j$.

As the notation suggests, $A_\phi \in \mathbb{R}_{\geq 0}^{d \times d}$ depends on all weights of all NNs. Moreover, it can effectively be interpreted as a weighted adjacency matrix similarly to what we presented in Section 2.3, since we have that

$$(A_\phi)_{ij} = 0 \implies \theta_{(j)} \text{ does not depend on variable } X_i \tag{8}$$

We note $\mathcal{G}_\phi$ to be the directed graph entailed by parameter $\phi$. We can now write our adapted acyclicity constraint:

$$h(\phi) \triangleq \operatorname{Tr} e^{A_\phi} - d = 0 \tag{9}$$

---

[1] Not all parameter vectors need to have the same dimensionality, but to simplify the notation, we suppose $m_j = m \;\; \forall j$

Note that we can compute the gradient of $h(\phi)$ w.r.t. $\phi$ (except at points of non-differentiability arising from the absolute value function, similar to standard neural networks with ReLU activations (Glorot et al., 2011); these points did not appear problematic in our experiments using SGD).

## 3.3 A DIFFERENTIABLE SCORE AND ITS OPTIMIZATION

We propose solving the maximum likelihood optimization problem

$$\max_{\phi} \mathbb{E}_{X \sim P_X} \sum_{j=1}^{d} \log p_j(X_j | X_{\pi_j^{\phi}}; \phi_{(j)}) \quad \text{s.t.} \quad \operatorname{Tr} e^{A_{\phi}} - d = 0 \tag{10}$$

where $\pi_j^{\phi}$ denotes the set of parents of node $j$ in graph $\mathcal{G}_{\phi}$. Note that $\sum_{j=1}^{d} \log p_j(X_j | X_{\pi_j^{\phi}}; \phi_{(j)})$ is a valid log-likelihood function when constraint (9) is satisfied.

As suggested in Zheng et al. (2018), we apply an augmented Lagrangian approach to get an approximate solution to program (10). Augmented Lagrangian methods consist of optimizing a sequence of subproblems for which the exact solutions are known to converge to a stationary point of the constrained problem under some regularity conditions (Bertsekas, 1999). In our case, each subproblem is

$$\max_{\phi} \mathcal{L}(\phi, \lambda_t, \mu_t) \triangleq \mathbb{E}_{X \sim P_X} \sum_{j=1}^{d} \log p_j(X_j | X_{\pi_j^{\phi}}; \phi_{(j)}) - \lambda_t h(\phi) - \frac{\mu_t}{2} h(\phi)^2 \tag{11}$$

where $\lambda_t$ and $\mu_t$ are the Lagrangian and penalty coefficients of the $t$th subproblem, respectively. These coefficients are updated after each subproblem is solved. Since GraN-DAG rests on neural networks, we propose to approximately solve each subproblem using a well-known stochastic gradient algorithm popular for NN in part for its implicit regularizing effect (Poggio et al., 2018). See Appendix A.1 for details regarding the optimization procedure.

In the current section, we presented GraN-DAG in a general manner without specifying explicitly which distribution family is parameterized by $\theta_{(j)}$. In principle, any distribution family could be employed as long as its log-likelihood can be computed and differentiated with respect to its parameter $\theta$. However, it is not always clear whether the exact solution of problem (10) recovers the ground truth graph $\mathcal{G}$. It will depend on both the modelling choice of GraN-DAG and the underlying CGM $(P_X, \mathcal{G})$.

**Proposition 1** *Let $\phi^*$ and $\mathcal{G}_{\phi^*}$ be the optimal solution to (10) and its corresponding graph, respectively. Let $\mathcal{M}(A)$ be the set of CGM $(P', \mathcal{G}')$ for which the assumptions in $A$ are satisfied and let $\mathcal{C}$ be the set of CGM $(P', \mathcal{G}')$ which can be represented by the model (e.g. NN outputting a Gaussian distribution). If the underlying CGM $(P_X, \mathcal{G}) \in \mathcal{C}$ and $\mathcal{C} = \mathcal{M}(A)$ for a specific set of assumptions $A$ such that $\mathcal{G}$ is identifiable from $P_X$, then $\mathcal{G}_{\phi^*} = \mathcal{G}$.*

*Proof:* Let $P_{\phi}$ be the joint distribution entailed by parameter $\phi$. Note that the population log-likelihood $\mathbb{E}_{X \sim P_X} \log p_{\phi}(X)$ is maximal iff $P_{\phi} = P_X$. We know this maximum can be achieved by a specific parameter $\phi^*$ since by hypothesis $(P_X, \mathcal{G}) \in \mathcal{C}$. Since $\mathcal{G}$ is identifiable from $P_X$, we know there exists no other CGM $(\tilde{P}_X, \tilde{\mathcal{G}}) \in \mathcal{C}$ such that $\tilde{\mathcal{G}} \neq \mathcal{G}$ and $\tilde{P}_X = P_X$. Hence $\mathcal{G}_{\phi^*}$ has to be equal to $\mathcal{G}$. ∎

In Section 4.1, we empirically explore the identifiable setting of nonlinear Gaussian ANMs introduced in Section 2.2. In practice, one should keep in mind that solving (10) exactly is hard since the problem is non-convex (the augmented Lagrangian converges only to a stationary point) and moreover we only have access to the empirical log-likelihood (Proposition 1 holds for the population case).

## 3.4 THRESHOLDING

The solution outputted by the augmented Lagrangian will satisfy the constraint only up to numerical precision, thus several entries of $A_{\phi}$ might not be exactly zero and require thresholding. To do so, we mask the inputs of each NN $j$ using a binary matrix $M_{(j)} \in \{0, 1\}^{d \times d}$ initialized to have $(M_{(j)})_{ii} = 1 \; \forall i \neq j$ and zeros everywhere else. Having $(M_{(j)})_{ii} = 0$ means the input $i$ of NN

$j$ has been thresholded. This mask is integrated in the product of Equation 6 by doing $C_{(j)} \triangleq |W_{(j)}^{(L+1)}| \dots |W_{(j)}^{(1)}| M_{(j)}$ without changing the interpretation of $C_{(j)}$ ($M_{(j)}$ can be seen simply as an extra layer in the NN). During optimization, if the entry $(A_\phi)_{ij}$ is smaller than the threshold $\epsilon = 10^{-4}$, the corresponding mask entry $(M_{(j)})_{ii}$ is set to zero, permanently. The masks $M_{(j)}$ are never updated via gradient descent. We also add an iterative thresholding step at the end to ensure the estimated graph $\mathcal{G}_\phi$ is acyclic (described in Appendix A.2).

## 3.5 Overfitting

In practice, we maximize an empirical estimate of the objective of problem (10). It is well known that this maximum likelihood score is prone to overfitting in the sense that adding edges can never reduce the maximal likelihood (Koller & Friedman, 2009). GraN-DAG gets around this issue in four ways. First, as we optimize a subproblem, we evaluate its objective on a held-out data set and declare convergence once it has stopped improving. This approach is known as *early stopping* (Prechelt, 1997). Second, to optimize (11), we use a stochastic gradient algorithm variant which is now known to have an implicit regularizing effect (Poggio et al., 2018). Third, once we have thresholded our graph estimate to be a DAG, we apply a final pruning step identical to what is done in CAM (Bühlmann et al., 2014) to remove spurious edges. This step performs a regression of each node against its parents and uses a significance test to decide which parents should be kept or not. Fourth, for graphs of 50 nodes or more, we apply a *preliminary neighbors selection* (PNS) before running the optimization procedure as was also recommended in Bühlmann et al. (2014). This procedure selects a set of potential parents for each variables. See Appendix A.3 for details on PNS and pruning. Many score-based approaches control overfitting by penalizing the number of edges in their score. For example, NOTEARS includes the L1 norm of its weighted adjacency matrix $U$ in its objective. GraN-DAG regularizes using PNS and pruning for ease of comparision to CAM, the most competitive approach in our experiments. The importance of PNS and pruning and their ability to reduce overfitting is illustrated in an ablation study presented in Appendix A.3. The study shows that PNS and pruning are both very important for the performance of GraN-DAG in terms of SHD, but do not have a significant effect in terms of SID. In these experiments, we also present NOTEARS and DAG-GNN with PNS and pruning, without noting a significant improvement.

## 3.6 Computational Complexity

To learn a graph, GraN-DAG relies on the proper training of neural networks on which it is built. We thus propose using a stochastic gradient method which is a standard choice when it comes to NN training because it scales well with both the sample size and the number of parameters and it implicitly regularizes learning. Similarly to NOTEARS, GraN-DAG requires the evaluation of the matrix exponential of $A_\phi$ at each iteration costing $\mathcal{O}(d^3)$. NOTEARS justifies the use of a batch proximal quasi-Newton algorithm by the low number of $\mathcal{O}(d^3)$ iterations required to converge. Since GraN-DAG uses a stochastic gradient method, one would expect it will require more iterations to converge. However, in practice we observe that GraN-DAG performs fewer iterations than NOTEARS before the augmented Lagrangian converges (see Table 4 of Appendix A.1). We hypothesize this is due to early stopping which avoids having to wait until the full convergence of the subproblems hence limiting the total number of iterations. Moreover, for the graph sizes considered in this paper ($d \leq 100$), the evaluation of $h(\phi)$ in GraN-DAG, which includes the matrix exponentiation, does not dominate the cost of each iteration ($\approx 4\%$ for 20 nodes and $\approx 13\%$ for 100 nodes graphs). Evaluating the approximate gradient of the log-likelihood (costing $\mathcal{O}(d^2)$ assuming a fixed minibatch size, NN depth and width) appears to be of greater importance for $d \leq 100$.

## 4 Experiments

In this section, we compare GraN-DAG to various baselines in the continuous paradigm, namely DAG-GNN (Yu et al., 2019) and NOTEARS (Zheng et al., 2018), and also in the combinatorial paradigm, namely CAM (Bühlmann et al., 2014), GSF (Huang et al., 2018), GES (Chickering, 2003) and PC (Spirtes et al., 2000). These methods are discussed in Section 5. In all experiments, each NN learned by GraN-DAG outputs the mean of a Gaussian distribution $\hat{\mu}_{(j)}$, i.e. $\theta_{(j)} := \hat{\mu}_{(j)}$ and $X_j | X_{\pi_j^{\mathcal{G}}} \sim \mathcal{N}(\hat{\mu}_{(j)}, \hat{\sigma}_{(j)}^2) \ \forall j$. The parameters $\hat{\sigma}_{(j)}^2$ are learned as well, but do not depend on

the parent variables $X_{\pi_j^{\mathcal{G}}}$ (unless otherwise stated). Note that this modelling choice matches the nonlinear Gaussian ANM introduced in Section 2.2.

We report the performance of random graphs sampled using the *Erdős-Rényi* (ER) scheme described in Appendix A.5 (denoted by RANDOM). For each approach, we evaluate the estimated graph on two metrics: the *structural hamming distance* (SHD) and the *structural interventional distance* (SID) (Peters & Bühlmann, 2013). The former simply counts the number of missing, falsely detected or reversed edges. The latter is especially well suited for causal inference since it counts the number of couples $(i,j)$ such that the interventional distribution $p(x_j|do(X_i = \bar{x}))$ would be miscalculated if we were to use the estimated graph to form the parent adjustement set. Note that GSF, GES and PC output only a CPDAG, hence the need to report a lower and an upper bound on the SID. See Appendix A.7 for more details on SHD and SID. All experiments were ran with publicly available code from the authors website. See Appendix A.8 for the details of their hyperparameters. In Appendix A.9, we explain how one could use a held-out data set to select the hyperparameters of score-based approaches and report the results of such a procedure on almost every settings discussed in the present section.

## 4.1 SYNTHETIC DATA

We have generated different *data set types* which vary along four dimensions: data generating process, number of nodes, level of edge sparsity and graph type. We consider two graph sampling schemes: *Erdős-Rényi* (ER) and *scale-free* (SF) (see Appendix A.5 for details). For each data set type, we sampled 10 data sets of 1000 examples as follows: First, a ground truth DAG $\mathcal{G}$ is randomly sampled following either the ER or the SF scheme. Then, the data is generated according to a specific sampling scheme.

The first data generating process we consider is the nonlinear Gaussian ANM (Gauss-ANM) introduced in Section 2.2 in which data is sampled following $X_j := f_j(X_{\pi_j^{\mathcal{G}}}) + N_j$ with mutually independent noises $N_j \sim \mathcal{N}(0, \sigma_j^2) \ \forall j$ where the functions $f_j$ are independently sampled from a Gaussian process with a unit bandwidth RBF kernel and with $\sigma_j^2$ sampled uniformly. As mentioned in Section 2.2, we know $\mathcal{G}$ to be identifiable from the distribution. Proposition 1 indicates that the modelling choice of GraN-DAG together with this synthetic data ensure that solving (10) to optimality would recover the correct graph. Note that NOTEARS and CAM also make the correct Gaussian noise assumption, but do not have enough capacity to represent the $f_j$ functions properly.

We considered graphs of 10, 20, 50 and 100 nodes. Tables 1 & 2 present results only for 10 and 50 nodes since the conclusions do not change with graphs of 20 or 100 nodes (see Appendix A.6 for these additional experiments). We consider graphs of $d$ and $4d$ edges (respectively denoted by ER1 and ER4 in the case of ER graphs). We report the performance of the popular GES and PC in Appendix A.6 since they are almost never on par with the best methods presented in this section.

Table 1: Results for ER and SF graphs of 10 nodes with Gauss-ANM data

| | ER1 | | ER4 | | SF1 | | SF4 | |
|---|---|---|---|---|---|---|---|---|
| | SHD | SID | SHD | SID | SHD | SID | SHD | SID |
| GraN-DAG | **1.7±2.5** | **1.7±3.1** | **8.3±2.8** | **21.8±8.9** | **1.2±1.1** | **4.1±6.1** | **9.9±4.0** | **16.4±6.0** |
| DAG-GNN | 11.4±3.1 | 37.6±14.4 | 35.1±1.5 | 81.9±4.7 | 9.9±1.1 | 29.7±15.8 | 20.8±1.9 | 48.4±15.6 |
| NOTEARS | 12.2±2.9 | 36.6±13.1 | 32.6±3.2 | 79.0±4.1 | 10.7±2.2 | 32.0±15.3 | 20.8±2.7 | 49.8±15.6 |
| CAM | **1.1±1.1** | **1.1±2.4** | 12.2±2.7 | 30.9±13.2 | **1.4±1.6** | **5.4±6.1** | **9.8±4.3** | **19.3±7.5** |
| GSF | 6.5±2.6 | [6.2±10.8 17.7±12.3] | 21.7±8.4 | [37.2±19.2 62.7±14.9] | **1.8±1.7** | [**2.0±5.1** **6.9±6.2**] | 8.5±4.2 | [**13.2±6.8** **20.6±12.1**] |
| RANDOM | 26.3±9.8 | 25.8±10.4 | 31.8±5.0 | 76.6±7.0 | 25.1±10.2 | 24.5±10.5 | 28.5±4.0 | 47.2±12.2 |

Table 2: Results for ER and SF graphs of 50 nodes with Gauss-ANM data

| | ER1 | | ER4 | | SF1 | | SF4 | |
|---|---|---|---|---|---|---|---|---|
| | SHD | SID | SHD | SID | SHD | SID | SHD | SID |
| GraN-DAG | **5.1±2.8** | **22.4±17.8** | **102.6±21.2** | **1060.1±109.4** | **25.5±6.2** | **90.0±18.9** | **111.3±12.3** | **271.2±65.4** |
| DAG-GNN | 49.2±7.9 | 304.4±105.1 | 191.9±15.2 | 2146.2±64 | 49.8±1.3 | 182.8±42.9 | 144.9±13.3 | 540.8±151.1 |
| NOTEARS | 62.8±9.2 | 327.3±119.9 | 202.3±14.3 | 2149.1±76.3 | 57.7±3.5 | 195.7±54.9 | 153.7±11.8 | 558.4±153.5 |
| CAM | **4.3±1.9** | **22.0±17.9** | 98.8±20.7 | 1197.2±125.9 | 24.1±6.2 | 85.7±31.9 | **111.2±13.3** | 320.7±152.6 |
| GSF | 25.6±5.1 | [21.1±23.1 79.2±33.5] | 81.8±18.8 | [906.6±214.7 1030.2±172.6] | 31.6±6.7 | [85.8±29.9 147.3±49.9] | 120.2±10.9 | [284.7±80.2 379.9±98.3] |
| RANDOM | 535.7±401.2 | 272.3±125.5 | 708.4±234.4 | 1921.3±203.5 | 514.0±360.0 | 381.3±190.3 | 660.6±194.9 | 1198.9±304.6 |

We now examine Tables 1 & 2 (the errors bars represent the standard deviation across datasets per task). We can see that, across all settings, GraN-DAG and CAM are the best performing methods, both in terms of SHD and SID, while GSF is not too far behind. The poor performance of NOTEARS can be explained by its inability to model nonlinear functions. In terms of SHD, DAG-GNN performs rarely better than NOTEARS while in terms of SID, it performs similarly to RANDOM in almost all cases except in scale-free networks of 50 nodes or more. Its poor performance might be due to its incorrect modelling assumptions and because its architecture uses a strong form of parameter sharing between the $f_j$ functions, which is not justified in a setup like ours. GSF performs always better than DAG-GNN and NOTEARS but performs as good as CAM and GraN-DAG only about half the time. Among the continuous approaches considered, GraN-DAG is the best performing on these synthetic tasks.

Even though CAM (wrongly) assumes that the functions $f_j$ are additive, i.e. $f_j(x_{\pi_j^{\mathcal{G}}}) = \sum_{i \in \pi_j^{\mathcal{G}}} f_{ij}(x_j) \; \forall j$, it manages to compete with GraN-DAG which does not make this incorrect modelling assumption[2]. This might partly be explained by a bias-variance trade-off. CAM is biased but has a lower variance than GraN-DAG due to its restricted capacity, resulting in both methods performing similarly. In Appendix A.4, we present an experiment showing that GraN-DAG can outperform CAM in higher sample size settings, suggesting this explanation is reasonable.

Having confirmed that GraN-DAG is competitive on the ideal Gauss-ANM data, we experimented with settings better adjusted to other models to see whether GraN-DAG remains competitive. We considered linear Gaussian data (better adjusted to NOTEARS) and nonlinear Gaussian data with additive functions (better adjusted to CAM) named LIN and ADD-FUNC, respectively. See Appendix A.5 for the details of their generation. We report the results of GraN-DAG and other baselines in Table 12 & 13 of the appendix. On linear Gaussian data, most methods score poorly in terms of SID which is probably due to the unidentifiability of the linear Gaussian model (when the noise variances are unequal). GraN-DAG and CAM perform similarly to NOTEARS in terms of SHD. On ADD-FUNC, CAM dominates all methods on most graph types considered (GraN-DAG is on par only for the 10 nodes ER1 graph). However, GraN-DAG outperforms all other methods which can be explained by the fact that the conditions of Proposition 1 are satisfied (supposing the functions $\sum_{i \in \pi_j^{\mathcal{G}}} f_{ij}(X_i)$ can be represented by the NNs).

We also considered synthetic data sets which do not satisfy the additive Gaussian noise assumption present in GraN-DAG, NOTEARS and CAM. We considered two kinds of *post nonlinear causal models* (Zhang & Hyvärinen, 2009), PNL-GP and PNL-MULT (see Appendix A.5 for details about their generation). A post nonlinear model has the form $X_j := g_j(f_j(X_{\pi_j^{\mathcal{G}}}) + N_j)$ where $N_j$ is a noise variable. Note that GraN-DAG (with the current modelling choice) and CAM do not have the representational power to express these conditional distributions, hence violating an assumption of Proposition 1. However, these data sets differ from the previous additive noise setup only by the nonlinearity $g_j$, hence offering a case of mild model misspecification. The results are reported in Table 14 of the appendix. GraN-DAG and CAM are outperforming DAG-GNN and NOTEARS except in SID for certain data sets where all methods score similarly to RANDOM. GraN-DAG and CAM have similar performance on all data sets except one where CAM is better. GSF performs worst than GraN-DAG (in both SHD and SID) on PNL-GP but not on PNL-MULT where it performs better in SID.

## 4.2 REAL AND PSEUDO-REAL DATA

We have tested all methods considered so far on a well known data set which measures the expression level of different proteins and phospholipids in human cells (Sachs et al., 2005). We trained only on the $n = 853$ observational samples. This dataset and its ground truth graph proposed in Sachs et al. (2005) (11 nodes and 17 edges) are often used in the probabilistic graphical model literature (Koller & Friedman, 2009). We also consider pseudo-real data sets sampled from the SynTReN generator (Van den Bulcke, 2006). This generator was designed to create synthetic transcriptional regulatory networks and produces simulated gene expression data that approximates experimental data. See Appendix A.5 for details of the generation.

---

[2]Although it is true that GraN-DAG does not wrongly assume that the functions $f_j$ are additive, it is not clear whether its neural networks can exactly represent functions sampled from the Gaussian process.

In applications, it is not clear whether the conditions of Proposition 1 hold since we do not know whether $(P_X, \mathcal{G}) \in \mathcal{C}$. This departure from identifiable settings is an occasion to explore a different modelling choice for GraN-DAG. In addition to the model presented at the beginning of this section, we consider an alternative, denoted GraN-DAG++, which allows the variance parameters $\hat{\sigma}^2_{(i)}$ to depend on the parent variables $X_{\pi_i^{\mathcal{G}}}$ through the NN, i.e. $\theta_{(i)} := (\hat{\mu}_{(i)}, \log \hat{\sigma}^2_{(i)})$. Note that this is violating the additive noise assumption (in ANMs, the noise is independent of the parent variables).

In addition to metrics used in Section 4.1, we also report SHD-C. To compute the SHD-C between two DAGs, we first map each of them to their corresponding CPDAG and measure the SHD between the two. This metric is useful to compare algorithms which only outputs a CPDAG like GSF, GES and PC to other methods which outputs a DAG. Results are reported in Table 3.

Table 3: Results on real and pseudo-real data

| | Protein signaling data set | | | SynTReN (20 nodes) | | |
| | SHD | SHD-C | SID | SHD | SHD-C | SID |
|---|---|---|---|---|---|---|
| GraN-DAG | 13 | 11 | 47 | **34.0±8.5** | **36.4±8.3** | 161.7±53.4 |
| GraN-DAG++ | 13 | 10 | 48 | **33.7±3.7** | 39.4±4.9 | 127.5±52.8 |
| DAG-GNN | 16 | 21 | 44 | 93.6±9.2 | 97.6±10.3 | 157.5±74.6 |
| NOTEARS | 21 | 21 | 44 | 151.8±28.2 | 156.1±28.7 | **110.7±66.7** |
| CAM | **12** | **9** | 55 | **40.5±6.8** | **41.4±7.1** | 152.3±48 |
| GSF | 18 | 10 | [44, 61] | 61.8±9.6 | 63.3±11.4 | **[76.7±51.1, 109.9±39.9]** |
| GES | 26 | 28 | **[34, 45]** | 82.6±9.3 | 85.6±10 | [157.2±48.3, 168.8±47.8] |
| PC | 17 | 11 | [47, 62] | **41.2±5.1** | **42.4±4.6** | [154.8±47.6, 179.3±55.6] |
| RANDOM | 21 | 20 | 60 | 84.7±53.8 | 86.7±55.8 | 175.8±64.7 |

First, all methods perform worse than what was reported for graphs of similar size in Section 4.1, both in terms of SID and SHD. This might be due to the lack of identifiability guarantees we face in applications. On the protein data set, GraN-DAG outperforms CAM in terms of SID (which differs from the general trend of Section 4.1) and arrive almost on par in terms of SHD and SHD-C. On this data set, DAG-GNN has a reasonable performance, beating GraN-DAG in SID, but not in SHD. On SynTReN, GraN-DAG obtains the best SHD but not the best SID. Overall, GraN-DAG is always competitive with the best methods of each task.

## 5 RELATED WORK

Most methods for structure learning from observational data make use of some identifiability results similar to the ones raised in Section 2.2. Roughly speaking, there are two classes of methods: *independence-based* and *score-based* methods. GraN-DAG falls into the second class.

Score-based methods (Koller & Friedman, 2009; Peters et al., 2017) cast the problem of structure learning as an optimization problem over the space of structures (DAGs or CPDAGs). Many popular algorithms tackle the combinatorial nature of the problem by performing a form of greedy search. GES (Chickering, 2003) is a popular example. It usually assumes a linear parametric model with Gaussian noise and greedily search the space of CPDAGs in order to optimize the Bayesian information criterion. GSF (Huang et al., 2018), is based on the same search algorithm as GES, but uses a generalized score function which can model nonlinear relationships. Other greedy approaches rely on parametric assumptions which render $\mathcal{G}$ fully identifiable. For example, Peters & Bühlman (2014) relies on a linear Gaussian model with equal variances to render the DAG identifiable. RE-SIT (Peters et al., 2014), assumes nonlinear relationships with additive Gaussian noise and greedily maximizes an independence-based score. However, RESIT does not scale well to graph of more than 20 nodes. CAM (Bühlmann et al., 2014) decouples the search for the optimal node ordering from the parents selection for each node and assumes an additive noise model (ANM) (Peters et al., 2017) in which the nonlinear functions are additive. As mentioned in Section 2.3, NOTEARS, proposed in Zheng et al. (2018), tackles the problem of finding an optimal DAG as a continuous constrained optimization program. This is a drastic departure from previous combinatorial approaches which enables the application of well studied numerical solvers for continuous optimizations. Recently, Yu et al. (2019) proposed DAG-GNN, a graph neural network architecture (GNN) which can be used to learn DAGs via the maximization of an evidence lower bound. By design, a GNN makes use of parameter sharing which we hypothesize is not well suited for most DAG learning tasks. To the best of our knowledge, DAG-GNN is the first approach extending the NOTEARS algorithm for structure

learning to support nonlinear relationships. Although Yu et al. (2019) provides empirical comparisons to linear approaches, namely NOTEARS and FGS (a faster extension of GES) (Ramsey et al., 2017), comparisons to greedy approaches supporting nonlinear relationships such as CAM and GSF are missing. Moreover, GraN-DAG significantly outperforms DAG-GNN on our benchmarks. There exists certain score-based approaches which uses integer linear programming (ILP) (Jaakkola et al., 2010; Cussens, 2011) which internally solve continuous linear relaxations. Connections between such methods and the continuous constrained approaches are yet to be explored.

When used with the additive Gaussian noise assumption, the theoretical guarantee of GraN-DAG rests on the identifiability of nonlinear Gaussian ANMs. Analogously to CAM and NOTEARS, this guarantee holds only if the correct identifiability assumptions hold in the data and if the score maximization problem is solved exactly (which is not the case in all three algorithms). DAG-GNN provides no theoretical justification for its approach. NOTEARS and CAM are designed to handle what is sometimes called the *high-dimensional setting* in which the number of samples is significantly smaller than the number of nodes. Bühlmann et al. (2014) provides consistency results for CAM in this setting. GraN-DAG and DAG-GNN were not designed with this setting in mind and would most likely fail if confronted to it. Solutions for fitting a neural network on less data points than input dimensions are not common in the NN literature.

Methods for causal discovery using NNs have already been proposed. SAM (Kalainathan et al., 2018) learns conditional NN generators using adversarial losses but does not enforce acyclicity. CGNN (Goudet et al., 2018), when used for multivariate data, requires an initial skeleton to learn the different functional relationships.

GraN-DAG has strong connections with MADE (Germain et al., 2015), a method used to learn distributions using a masked NN which enforces the so-called *autoregressive property*. The autoregressive property and acyclicity are in fact equivalent. MADE does not learn the weight masking, it fixes it at the beginning of the procedure. GraN-DAG could be used with a unique NN taking as input all variables and outputting parameters for all conditional distributions. In this case, it would be similar to MADE, except the variable ordering would be learned from data instead of fixed *a priori*.

## 6 CONCLUSION

The continuous constrained approach to structure learning has the advantage of being more global than other approximate greedy methods (since it updates all edges at each step based on the gradient of the score but also the acyclicity constraint) and allows to replace task-specific greedy algorithms by appropriate off-the-shelf numerical solvers. In this work, we have introduced GraN-DAG, a novel score-based approach for structure learning supporting nonlinear relationships while leveraging a continuous optimization paradigm. The method rests on a novel characterization of acyclicity for NNs based on the work of Zheng et al. (2018). We showed GraN-DAG outperforms other gradient-based approaches, namely NOTEARS and its recent nonlinear extension DAG-GNN, on the synthetic data sets considered in Section 4.1 while being competitive on real and pseudo-real data sets of Section 4.2. Compared to greedy approaches, GraN-DAG is competitive across all datasets considered. To the best of our knowledge, GraN-DAG is the first approach leveraging the continuous paradigm introduced in Zheng et al. (2018) which has been shown to be competitive with state of the art methods supporting nonlinear relationships.

### ACKNOWLEDGMENTS

This research was partially supported by the Canada CIFAR AI Chair Program and by a Google Focused Research award. The authors would like to thank Rémi Le Priol, Tatjana Chavdarova, Charles Guille-Escuret, Nicolas Gagné and Yoshua Bengio for insightful discussions as well as Alexandre Drouin and Florian Bordes for technical support. The experiments were in part enabled by computational resources provided by Calcul Québec, Compute Canada and Element AI.

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

# A APPENDIX

## A.1 OPTIMIZATION

Let us recall the augmented Lagrangian:

$$\max_{\phi} \mathcal{L}(\phi, \lambda_t, \mu_t) \triangleq \mathbb{E}_{X \sim P_X} \sum_{i=1}^{d} \log p_i(X_i | X_{\pi_i^{\phi}}; \phi_{(i)}) - \lambda_t h(\phi) - \frac{\mu_t}{2} h(\phi)^2 \qquad (12)$$

where $\lambda_t$ and $\mu_t$ are the Lagrangian and penalty coefficients of the $t$th subproblem, respectively. In all our experiments, we initialize those coefficients using $\lambda_0 = 0$ and $\mu_0 = 10^{-3}$. We approximately solve each non-convex subproblem using RMSprop (Tieleman & Hinton, 2012), a stochastic gradient descent variant popular for NNs. We use the following gradient estimate:

$$\nabla_{\phi} \mathcal{L}(\phi, \lambda_t, \mu_t) \approx \nabla_{\phi} \hat{\mathcal{L}}_B(\phi, \lambda_t, \mu_t)$$

$$\text{with } \hat{\mathcal{L}}_B(\phi, \lambda_t, \mu_t) \triangleq \frac{1}{|B|} \sum_{x \in B} \sum_{i=1}^{d} \log p_i(x_i | x_{\pi_i^{\phi}}; \phi_{(i)}) - \lambda_t h(\phi) - \frac{\mu_t}{2} h(\phi)^2 \qquad (13)$$

where $B$ is a minibatch sampled from the data set and $|B|$ is the minibatch size. The gradient estimate $\nabla_{\phi} \hat{\mathcal{L}}_B(\phi, \lambda_t, \mu_t)$ can be computed using standard deep learning libraries. We consider a subproblem has converged when $\hat{\mathcal{L}}_H(\phi, \lambda_t, \mu_t)$ evaluated on a held-out data set $H$ stops increasing. Let $\phi_t^*$ be the approximate solution to subproblem $t$. Then, $\lambda_t$ and $\mu_t$ are updated according to the following rule:

$$
\begin{aligned}
\lambda_{t+1} &\leftarrow \lambda_t + \mu_t h\left(\phi_t^*\right) \\
\mu_{t+1} &\leftarrow \begin{cases} \eta \mu_t, & \text{if } h\left(\phi_t^*\right) > \gamma h\left(\phi_{t-1}^*\right) \\ \mu_t, & \text{otherwise} \end{cases}
\end{aligned}
\qquad (14)
$$

with $\eta = 10$ and $\gamma = 0.9$. Each subproblem $t$ is initialized using the previous subproblem solution $\phi_{t-1}^*$. The augmented Lagrangian method stops when $h(\phi) \leq 10^{-8}$.

**Total number of iterations before augmented Lagrangian converges:** In GraN-DAG and NOTEARS, every subproblem is approximately solved using an iterative algorithm. Let $T$ be the number of subproblems solved before the convergence of the augmented Lagrangian. For a given subproblem $t$, let $K_t$ be the number of iterations executed to approximately solve it. Let $I = \sum_{t=1}^{T} K_t$ be the *total number of iterations* before the augmented Lagrangian converges. Table 4 reports the total number of iterations $I$ for GraN-DAG and NOTEARS, averaged over ten data sets. Note that the matrix exponential is evaluated once per iteration. Even though GraN-DAG uses a stochastic gradient algorithm, it requires less iterations than NOTEARS which uses a batch proximal quasi-Newton method. We hypothesize early stopping avoids having to wait until full convergence before moving to the next subproblem, hence reducing the total number of iterations. Note that GraN-DAG total run time is still larger than NOTEARS due to its gradient requiring more computation to evaluate (total runtime $\approx 10$ minutes against $\approx 1$ minute for 20 nodes graphs and $\approx 4$ hours against $\approx 1$ hour for 100 nodes graphs). GraN-DAG runtime on 100 nodes graphs can be roughly halved when executed on GPU.

Table 4: Total number of iterations ($\times 10^3$) before augmented Lagrangian converges on Gauss-ANM data.

|  | 20 nodes ER1 | 20 nodes ER4 | 100 nodes ER1 | 100 nodes ER4 |
|---|---|---|---|---|
| GraN-DAG | $27.3 \pm 3.6$ | $30.4 \pm 4.2$ | $23.1 \pm 0.7$ | $23.1 \pm 0.8$ |
| NOTEARS | $67.1 \pm 35.3$ | $72.3 \pm 24.3$ | $243.6 \pm 12.3$ | $232.4 \pm 12.9$ |

## A.2 THRESHOLDING TO ENSURE ACYCLICITY

The augmented Lagrangian outputs $\phi_T^*$ where $T$ is the number of subproblems solved before declaring convergence. Note that the weighted adjacency matrix $A_{\phi_T^*}$ will most likely not represent an acyclic graph, even if we threshold as we learn, as explained in Section 3.4. We need to remove

additional edges to obtain a DAG (edges are removed using the mask presented in Section 3.4). One option would be to remove edges one by one until a DAG is obtained, starting from the edge $(i, j)$ with the lowest $(A_{\phi_T^*})_{ij}$ up to the edge with the highest $(A_{\phi_T^*})_{ij}$. This amounts to gradually increasing the threshold $\epsilon$ until $A_{\phi_T^*}$ is acyclic. However, this approach has the following flaw: It is possible to have $(A_{\phi_T^*})_{ij}$ significantly higher than zero while having $\theta_{(j)}$ almost completely independent of variable $X_i$. This can happen for at least two reasons. First, the NN paths from input $i$ to output $k$ might end up cancelling each others, rendering the input $i$ inactive. Second, some neurons of the NNs might always be saturated for the observed range of inputs, rendering some NN paths *effectively inactive* without being inactive in the sense described in Section 3.1. Those two observations illustrate the fact that having $(A_{\phi_T^*})_{ij} = 0$ is only a sufficient condition to have $\theta_{(j)}$ independent of variable $X_i$ and not a necessary one.

To avoid this issue, we consider the following alternative. Consider the function $\mathcal{L} : \mathbb{R}^d \mapsto \mathbb{R}^d$ which maps all $d$ variables to their respective conditional likelihoods, i.e. $\mathcal{L}_i(X) \triangleq p_i(X_i \mid X_{\pi_i^{\phi_T^*}}) \; \forall i$.

We consider the following expected Jacobian matrix

$$\mathcal{J} \triangleq \mathbb{E}_{X \sim P_X} \left| \frac{\partial \mathcal{L}}{\partial X} \right|^\top \tag{15}$$

where $\left| \frac{\partial \mathcal{L}}{\partial X} \right|$ is the Jacobian matrix of $\mathcal{L}$ evaluated at $X$, in absolute value (element-wise). Similarly to $(A_{\phi_T^*})_{ij}$, the entry $\mathcal{J}_{ij}$ can be loosely interpreted as the strength of edge $(i, j)$. We propose removing edges starting from the lowest $\mathcal{J}_{ij}$ to the highest, stopping as soon as acyclicity is achieved. We believe $\mathcal{J}$ is better than $A_{\phi_T^*}$ at capturing which NN inputs are effectively inactive since it takes into account NN paths cancelling each others and saturated neurons. Empirically, we found that using $\mathcal{J}$ instead of $A_{\phi_T^*}$ yields better results, and thus we report the results with $\mathcal{J}$ in this paper.

### A.3 PRELIMINARY NEIGHBORHOOD SELECTION AND DAG PRUNING

**PNS:** For graphs of 50 nodes or more, GraN-DAG performs a *preliminary neighborhood selection* (PNS) similar to what has been proposed in Bühlmann et al. (2014). This procedure applies a variable selection method to get a set of possible parents for each node. This is done by fitting an *extremely randomized trees* (Geurts et al., 2006) (using `ExtraTreesRegressor` from `scikit-learn`) for each variable against all the other variables. For each node a feature importance score based on the gain of purity is calculated. Only nodes that have a feature importance score higher than $0.75 \cdot$ `mean` are kept as potential parent, where `mean` is the mean of the feature importance scores of all nodes. Although the use of PNS in CAM was motivated by gains in computation time, GraN-DAG uses it to avoid overfitting, without reducing the computation time.

**Pruning:** Once the thresholding is performed and a DAG is obtained as described in A.2, GraN-DAG performs a pruning step identical to CAM (Bühlmann et al., 2014) in order to remove spurious edges. We use the implementation of Bühlmann et al. (2014) based on the R function `gamboost` from the `mboost` package. For each variable $X_i$, a generalized additive model is fitted against the current parents of $X_i$ and a significance test of covariates is applied. Parents with a p-value higher than 0.001 are removed from the parent set. Similarly to what Bühlmann et al. (2014) observed, this pruning phase generally has the effect of greatly reducing the SHD without considerably changing the SID.

**Ablation study:** In Table 5, we present an ablation study which shows the effect of adding PNS and pruning to GraN-DAG on different performance metrics and on the negative log-likelihood (NLL) of the training and validation set. Note that, before computing both NLL, we reset all parameters of GraN-DAG except the mask and retrained the model on the training set without any acyclicity constraint (acyclicity is already ensure by the masks at this point). This retraining procedure is important since the pruning removes edges (i.e. some additional NN inputs are masked) and it affects the likelihood of the model (hence the need to retrain).

Table 5: PNS and pruning ablation study for GraN-DAG (averaged over 10 datasets from ER1 with 50 nodes)

| PNS | Pruning | SHD | SID | NLL (train) | NLL (validation) |
|-----|---------|-----|-----|-------------|------------------|
| False | False | 1086.8±48.8 | 31.6±23.6 | 0.36±0.07 | 1.44±0.21 |
| True | False | 540.4±70.3 | 17.4±16.7 | 0.52±0.08 | 1.16±0.17 |
| False | True | 11.8±5.0 | 39.7±25.5 | 0.78±0.12 | 0.84±0.12 |
| True | True | 6.1±3.3 | 29.3±19.5 | 0.78±0.13 | 0.83±0.12 |

A first observation is that adding PNS and pruning improve the NLL on the validation set while deteriorating the NLL on the training set, showing that those two steps are indeed reducing overfitting. Secondly, the effect on SHD is really important while the effect on SID is almost nonexistent. This can be explained by the fact that SID has more to do with the ordering of the nodes than with false positive edges. For instance, if we have a complete DAG with a node ordering coherent with the ground truth graph, the SID is zero, but the SHD is not due to all the false positive edges. Without the regularizing effect of PNS and pruning, GraN-DAG manages to find a DAG with a good ordering but with many spurious edges (explaining the poor SHD, the good SID and the big gap between the NLL of the training set and validation set). PNS and pruning helps reducing the number of spurious edges, hence improving SHD.

We also implemented PNS and pruning for NOTEARS and DAG-GNN to see whether their performance could also be improved. Table 6 reports an ablation study for DAG-GNN and NOTEARS. First, the SHD improvement is not as important as for GraN-DAG and is almost not statistically significant. The improved SHD does not come close to performance of GraN-DAG. Second, PNS and pruning do not have a significant effect of SID, as was the case for GraN-DAG. The lack of improvement for those methods is probably due to the fact that they are not overfitting like GraN-DAG, as the training and validation (unregularized) scores shows. NOTEARS captures only linear relationships, thus it will have a hard time overfitting nonlinear data and DAG-GNN uses a strong form of parameter sharing between its conditional densities which possibly cause underfitting in a setup where all the parameters of the conditionals are sampled independently.

Moreover, DAG-GNN and NOTEARS threshold aggressively their respective weighted adjacency matrix at the end of training (with the default parameters used in the code), which also acts as a form of heavy regularization, and allow them to remove many spurious edges. GraN-DAG without PNS and pruning does not threshold as strongly by default which explains the high SHD of Table 5. To test this explanation, we removed all edges $(i, j)$ for which $(A_\phi)_{ij} < 0.3$[3] for GraN-DAG and obtained an SHD of 29.4±15.9 and an SID of 85.6±45.7, showing a significant improvement over NOTEARS and DAG-GNN, even without PNS and pruning.

Table 6: PNS and pruning ablation study for DAG-GNN and NOTEARS (averaged over 10 datasets from ER1 with 50 nodes)

| Algorithm | PNS | Pruning | SHD | SID | Score (train) | Score (validation) |
|-----------|-----|---------|-----|-----|---------------|-------------------|
| DAG-GNN | False | False | 56.8±11.1 | 322.9±103.8 | -2.8±1.5 | -2.2±1.6 |
| | True | False | 55.5±10.2 | 314.5±107.6 | -2.1±1.6 | -2.1±1.7 |
| | False | True | 49.4±7.8 | 325.1±103.7 | -1.8±1.1 | -1.8±1.2 |
| | True | True | 47.7±7.3 | 316.5±105.6 | -1.9±1.6 | -1.9±1.6 |
| NOTEARS | False | False | 64.2±9.5 | 327.1±110.9 | -23.1±1.8 | -23.2±2.1 |
| | True | False | 54.1±10.9 | 321.5±104.5 | -25.2±2.7 | -25.4±2.8 |
| | False | True | 49.5±8.8 | 327.7±111.3 | -26.7±2.0 | -26.8±2.1 |
| | True | True | 49.0±7.6 | 326.4±106.9 | -26.23±2.2 | -26.4±2.4 |

---

[3]This was the default value of thresholding used in NOTEARS and DAG-GNN.

A.4 LARGE SAMPLE SIZE EXPERIMENT

In this section, we test the bias-variance hypothesis which attempts to explain why CAM is on par with GraN-DAG on Gauss-ANM data even if its model wrongly assumes that the $f_j$ functions are additive. Table 7 reports the performance of GraN-DAG and CAM for different sample sizes. We can see that, as the sample size grows, GraN-DAG ends up outperforming CAM in terms of SID while staying on par in terms of SHD. We explain this observation by the fact that a larger sample size reduces variance for GraN-DAG thus allowing it to leverage its greater capacity against CAM which is stuck with its modelling bias. Both algorithms were run with their respective default hyperparameter combination.

This experiment suggests GraN-DAG could be an appealing option in settings where the sample size is substantial. The present paper focuses on sample sizes typically encountered in the structure/causal learning litterature and leave this question for future work.

Table 7: Effect of sample size - Gauss-ANM 50 nodes ER4 (averaged over 10 datasets)

| Sample size | Method | SHD | SID |
|---|---|---|---|
| 500 | CAM | $123.5 \pm 13.9$ | $1181.2 \pm 160.8$ |
| | GraN-DAG | $130.2 \pm 14.4$ | $1246.4 \pm 126.1$ |
| 1000 | CAM | $103.7 \pm 15.2$ | $1074.7 \pm 125.8$ |
| | GraN-DAG | $104.4 \pm 15.3$ | $942.1 \pm 69.8$ |
| 5000 | CAM | $74.1 \pm 13.2$ | $845.0 \pm 159.8$ |
| | GraN-DAG | $71.9 \pm 15.9$ | $554.1 \pm 117.9$ |
| 10000 | CAM | $66.3 \pm 16.0$ | $808.1 \pm 142.9$ |
| | GraN-DAG | $65.9 \pm 19.8$ | $453.4 \pm 171.7$ |

A.5 DETAILS ON DATA SETS GENERATION

**Synthetic data sets:** For each data set type, 10 data sets are sampled with 1000 examples each. As the synthetic data introduced in Section 4.1, for each data set, a ground truth DAG $\mathcal{G}$ is randomly sampled following the ER scheme and then the data is generated. Unless otherwise stated, all root variables are sampled from $\mathcal{U}[-1, 1]$.

- *Gauss-ANM* is generated following $X_j := f_j(X_{\pi_j^{\mathcal{G}}}) + N_j \ \forall j$ with mutually independent noises $N_j \sim \mathcal{N}(0, \sigma_j^2) \ \forall j$ where the functions $f_j$ are independently sampled from a Gaussian process with a unit bandwidth RBF kernel and $\sigma_j^2 \sim \mathcal{U}[0.4, 0.8]$. Source nodes are Gaussian with zero mean and variance sampled from $\mathcal{U}[1, 2]$

- *LIN* is generated following $X_j | X_{\pi_j^{\mathcal{G}}} \sim w_j^T X_{\pi_j^{\mathcal{G}}} + 0.2 \cdot \mathcal{N}(0, \sigma_j^2) \ \forall j$ where $\sigma_j^2 \sim \mathcal{U}[1, 2]$ and $w_j$ is a vector of $|\pi_j^{\mathcal{G}}|$ coefficients each sampled from $\mathcal{U}[0, 1]$.

- *ADD-FUNC* is generated following $X_j | X_{\pi_j^{\mathcal{G}}} \sim \sum_{i \in \pi_j^{\mathcal{G}}} f_{j,i}(X_i) + 0.2 \cdot \mathcal{N}(0, \sigma_j^2) \ \forall j$ where $\sigma_j^2 \sim \mathcal{U}[1, 2]$ and the functions $f_{j,i}$ are independently sampled from a Gaussian process with bandwidth one. This model is adapted from Bühlmann et al. (2014).

- *PNL-GP* is generated following $X_j | X_{\pi_j^{\mathcal{G}}} \sim \sigma(f_j(X_{\pi_j^{\mathcal{G}}}) + Laplace(0, l_j)) \ \forall j$ with the functions $f_j$ independently sampled from a Gaussian process with bandwidth one and $l_j \sim \mathcal{U}[0, 1]$. In the two-variable case, this model is identifiable following the Corollary 9 from Zhang & Hyvärinen (2009). To get identifiability according to this corollary, it is important to use non-Gaussian noise, explaining our design choices.

- *PNL-MULT* is generated following $X_j | X_{\pi_j^{\mathcal{G}}} \sim \exp(\log(\sum_{i \in \pi_j^{\mathcal{G}}} X_i) + |\mathcal{N}(0, \sigma_j^2)|) \ \forall j$ where $\sigma_j^2 \sim \mathcal{U}[0, 1]$. Root variables are sampled from $\mathcal{U}[0, 2]$. This model is adapted from Zhang et al. (2015).

**SynTReN:** Ten datasets have been generated using the SynTReN generator (http://bioinformatics.intec.ugent.be/kmarchal/SynTReN/index.html) using the

software default parameters except for the *probability for complex 2-regulator interactions* that was set to 1 and the random seeds used were 0 to 9. Each dataset contains 500 samples and comes from a 20 nodes graph.

**Graph types:** *Erdős-Rényi* (ER) graphs are generated by randomly sampling a topological order and by adding directed edges were it is allowed independently with probability $p = \frac{2e}{d^2-d}$ were $e$ is the expected number of edges in the resulting DAG. *Scale-free* (SF) graphs were generated using the Barabási-Albert model (Barabási & Albert, 1999) which is based on preferential attachment. Nodes are added one by one. Between the new node and the existing nodes, $m$ edges (where $m$ is equal to $d$ or $4d$) will be added. An existing node $i$ have the probability $p(k_i) = \frac{k_i}{\sum_j k_j}$ to be chosen, where $k_i$ represents the degree of the node $i$. While ER graphs have a degree distribution following a Poisson distribution, SF graphs have a degree distribution following a power law: few nodes, often called *hubs*, have a high degree. Barabási (2009) have stated that these types of graphs have similar properties to real-world networks which can be found in many different fields, although these claims remain controversial (Clauset et al., 2009).

## A.6 SUPPLEMENTARY EXPERIMENTS

**Gauss-ANM:** The results for 20 and 100 nodes are presented in Table 8 and 9 using the same Gauss-ANM data set types introduced in Section 4.1. The conclusions drawn remains similar to the 10 and 50 nodes experiments. For GES and PC, the SHD and SID are respectively presented in Table 10 and 11. Their performances do not compare favorably to the GraN-DAG nor CAM. Figure 1 shows the entries of the weighted adjacency matrix $A_\phi$ as training proceeds in a typical run for 10 nodes.

**LIN & ADD-FUNC:** Experiments with LIN and ADD-FUNC data is reported in Table 12 & 13. The details of their generation are given in Appendix A.5.

**PNL-GP & PNL-MULT:** Table 14 contains the performance of GraN-DAG and other baselines on post nonlinear data discussed in Section 4.1.

Table 8: Results for ER and SF graphs of 20 nodes with Gauss-ANM data

| | ER1 | | ER4 | | SF1 | | SF4 | |
|---|---|---|---|---|---|---|---|---|
| | SHD | SID | SHD | SID | SHD | SID | SHD | SID |
| GraN-DAG | **4.0** ±**3.4** | **17.9**±**19.5** | 45.2±10.7 | **165.1**±**21.0** | 7.6±2.5 | 28.8±10.4 | 36.8±5.1 | 62.5±18.8 |
| DAG-GNN | 25.6±7.5 | 109.1±53.1 | 75.0±7.7 | 344.8±17.0 | 19.5±1.8 | 60.1±12.8 | 49.5±5.4 | 115.2±33.3 |
| NOTEARS | 30.3±7.8 | 107.3±47.6 | 79.0±8.0 | 346.6±13.2 | 23.9±3.5 | 69.4±19.7 | 52.0±4.5 | 120.5±32.5 |
| CAM | **2.7**±**1.8** | **10.6**±**8.6** | **41.0**±**11.9** | 157.9±41.2 | **5.7**±**2.6** | **23.3**±**18.0** | **35.6**±**4.5** | **59.1**±**18.8** |
| GSF | 12.3±4.6 | [15.0±19.9 45.6±22.9] | **41.8**±**13.8** | [153.7±49.4 201.6±37.9] | 7.4±3.5 | [5.7±7.1 27.3±13.2] | 38.6±3.6 | [54.9±14.4 86.7±24.2] |
| RANDOM | 103.0±39.6 | 94.3±53.0 | 117.5±25.9 | 298.5±28.7 | 105.2±48.8 | 81.1±54.4 | 121.5±28.5 | 204.8±38.5 |

Table 9: Results for ER and SF graphs of 100 nodes with Gauss-ANM data

| | ER1 | | ER4 | | SF1 | | SF4 | |
|---|---|---|---|---|---|---|---|---|
| | SHD | SID | SHD | SID | SHD | SID | SHD | SID |
| GraN-DAG | **15.1**±**6.0** | **83.9**±**46.0** | 206.6±31.5 | 4207.3±419.7 | 59.2±7.7 | 265.4±64.2 | 262.7±19.6 | 872.0±130.4 |
| DAG-GNN | 110.2±10.5 | 883.0±320.9 | 379.5±24.7 | 8036.1±656.2 | 97.6±1.5 | 438.6±112.7 | 316.0±14.3 | 1394.6±165.9 |
| NOTEARS | 125.6±12.1 | 913.1±343.8 | 387.8±25.3 | 8124.7±577.4 | 111.7±5.4 | 484.3±138.4 | 327.2±15.8 | 1442.8±210.1 |
| CAM | **17.3**±**4.5** | **124.9**±**65.0** | **186.4**±**28.8** | 4601.9±482.7 | 52.7±9.3 | 230.3±36.9 | 255.6±21.7 | 845.8±161.3 |
| GSF | 66.8±7.3 | [104.7±59.5 238.6±59.3] | > 12 hours[4] | — | 71.4±11.2 | [212.7±71.1 325.3±105.2] | 275.9±21.0 | [793.9±152.5 993.4±149.2] |
| RANDOM | 1561.6±1133.4 | 1175.3±547.9 | 2380.9±1458.0 | 7729.7±1056.0 | 2222.2±1141.2 | 1164.2±593.3 | 2485.0±1403.9 | 4206.4±1642.1 |

---

[4]Note that GSF results are missing for two data set types in Tables 9 and 14. This is because the search algorithm could not finish within 12 hours, even when the maximal in-degree was limited to 5. All other methods could run in less than 6 hours.

Table 10: SHD for GES and PC (against GraN-DAG for reference) with Gauss-ANM data

|  | 10 nodes | | 20 nodes | | 50 nodes | | 100 nodes | |
|---|---|---|---|---|---|---|---|---|
|  | ER1 | ER4 | ER1 | ER4 | ER1 | ER4 | ER1 | ER4 |
| GraN-DAG | 1.7±2.5 | 8.3±2.8 | 4.0 ±3.4 | 45.2±10.7 | 5.1±2.8 | 102.6±21.2 | 15.1±6.0 | 206.6±31.5 |
| GES | 13.8±4.8 | 32.3±4.3 | 43.3±12.4 | 94.6±9.8 | 106.6±24.7 | 254.4±39.3 | 292.9±33.6 | 542.6±51.2 |
| PC | 8.4±3 | 34±2.6 | 20.136.4±6.5 | 73.1±5.8 | 44.0±11.6 | 183.6±20 | 95.2±9.1 | 358.8±20.6 |
|  | SF1 | SF4 | SF1 | SF4 | SF1 | SF4 | SF1 | SF4 |
| GraN-DAG | 1.2±1.1 | 9.9±4.0 | 7.6±2.5 | 36.8±5.1 | 25.5±6.2 | 111.3±12.3 | 59.2±7.7 | 262.7±19.6 |
| GES | 8.1±2.4 | 17.4±4.5 | 26.2±7.5 | 50.7±6.2 | 73.9±7.4 | 178.8±16.5 | 190.3±22 | 408.7±24.9 |
| PC | 4.8±2.4 | 16.4±2.8 | 13.6±2.1 | 44.4±4.6 | 43.1±5.7 | 135.4±10.7 | 97.6±6.6 | 314.2±17.5 |

Table 11: Lower and upper bound on the SID for GES and PC (against GraN-DAG for reference) with Gauss-ANM data. See Appendix A.7 for details on how to compute SID for CPDAGs.

|  | 10 nodes | | 20 nodes | | 50 nodes | | 100 nodes | |
|---|---|---|---|---|---|---|---|---|
|  | ER1 | ER4 | ER1 | ER4 | ER1 | ER4 | ER1 | ER4 |
| GraN-DAG | 1.7±3.1 | 21.8±8.9 | 17.9±19.5 | 165.1±21.0 | 22.4±17.8 | 1060.1±109.4 | 83.9±46.0 | 4207.3±419.7 |
| GES | [24.1±17.3 27.2±17.5] | [ 68.5±10.5 75±7] | [ 62.1±44 65.7±44.5] | [ 301.9±19.4 319.3±13.3] | [150.9±72.7 155.1±74] | [ 1996.6±73.1 2032.9±88.7] | [ 582.5±391.1 598.9±408.6] | [ 8054±524.8 8124.2±470.2] |
| PC | [22.6±15.5 27.3±13.1] | [78.1±7.4 79.2±5.7] | [80.9±51.1 94.9±46.1] | [316.7±25.7 328.7±25.6] | [222.7±138 256.7±127.3] | [2167.9±88.4 2178.8±80.8] | [620.7±240.9 702.5±255.8] | [8236.9±478.5 8265.4±470.2] |
|  | SF1 | SF4 | SF1 | SF4 | SF1 | SF4 | SF1 | SF4 |
| GraN-DAG | 4.1±6.1 | 16.4±6.0 | 28.8±10.4 | 62.5±18.8 | 90.0±18.9 | 271.2±65.4 | 265.4±64.2 | 872.0±130.4 |
| GES | [11.6±9.2 16.4±11.7] | [39.3±11.2 43.9±14.9] | [54.9±23.1 57.9±24.6] | [89.5±38.4 105.1±44.3] | [171.6±70.1 182.7±77] | [496.3±154.1 529.7±184.5] | [511.5±257.6 524±252.2] | [1421.7±247.4 1485.4±233.6] |
| PC | [8.3±4.6 16.8±12.3] | [36.5±6.2 41.7±6.9] | [42.2±14 59.7±14.9] | [95.6±37 118.5±30] | [124.2±38.3 167.1±41.4] | [453.2±115.9 538±143.7] | [414.5±124.4 486.5±120.9] | [1369.2±259.9 1513.7±296.2] |

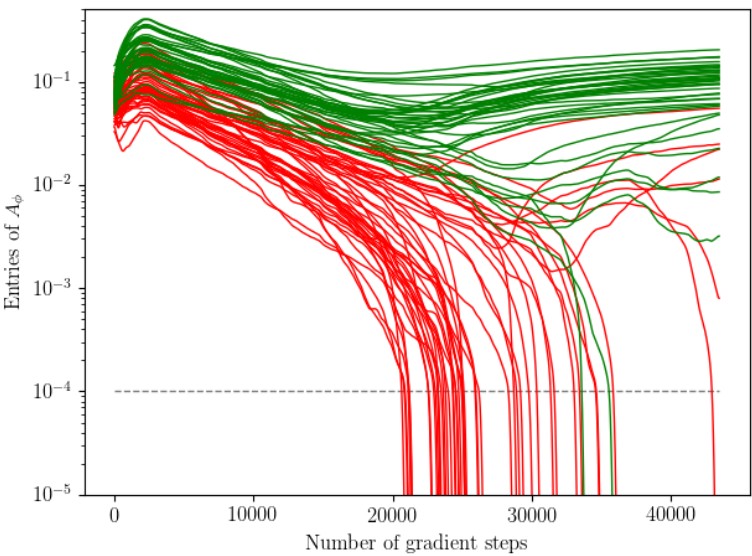

Figure 1: Entries of the weighted adjacency matrix $A_\phi$ as training proceeds in GraN-DAG for a synthetic data set ER4 with 10 nodes. Green curves represent edges which appear in the ground truth graph while red ones represent edges which do not. The horizontal dashed line at $10^{-4}$ is the threshold $\epsilon$ introduced in Section 3.4. We can see that GraN-DAG successfully recovers most edges correctly while keeping few spurious edges.

Table 12: LIN

| #Nodes | 10 | | | | 50 | | | |
|---|---|---|---|---|---|---|---|---|
| Graph Type | ER1 | | ER4 | | ER1 | | ER4 | |
| Metrics | SHD | SID | SHD | SID | SHD | SID | SHD | SID |
| Method | | | | | | | | |
| GraN-DAG | **7.2 ± 2.0** | 27.3 ± 8.1 | 30.7 ± 3.3 | 75.8 ± 6.9 | **33.9 ± 8.6** | **255.8 ± 158.4** | 181.9 ± 24.0 | 2035.8 ± 137.2 |
| DAG-GNN | 10.3 ± 3.5 | 39.6 ± 14.7 | **18.9 ± 4.8** | **63.7 ± 8.9** | 54.1 ± 9.2 | 330.4 ± 117.1 | **130.3 ± 17.3** | **1937.5 ± 89.8** |
| NOTEARS | 9.0 ± 3.0 | 35.3 ± 13.4 | 27.9 ± 4.3 | **72.1 ± 7.9** | 45.5 ± 7.8 | 310.7 ± 125.9 | **126.1 ± 13.0** | 1971.1 ± 134.3 |
| CAM | 10.2 ± 6.3 | 31.2 ± 10.9 | 33.6 ± 3.3 | 77.5 ± 2.3 | **36.2 ± 5.8** | **234.8 ± 105.1** | 182.5 ± 17.6 | 1948.7 ± 113.5 |
| GSF | 9.2 ± 2.9 | [19.5 ± 14.6 31.6 ± 17.3] | 38.6 ± 3.7 | [73.8 ± 7.6 85.2 ± 8.3] | 46.7 ± 4.1 | [**176.4 ± 98.8** 215.0 ± 108.9] | > 12 hours | |
| RANDOM | 22.0 ± 2.9 | 30.0 ± 13.8 | 34.4 ± 2.4 | 78.8 ± 5.5 | 692.6 ± 7.5 | 360.3 ± 141.4 | 715.9 ± 16.0 | **1932.7 ± 40.2** |

Table 13: ADD-FUNC

| #Nodes | 10 | | | | 50 | | | |
|---|---|---|---|---|---|---|---|---|
| Graph Type | ER1 | | ER4 | | ER1 | | ER4 | |
| Metrics | SHD | SID | SHD | SID | SHD | SID | SHD | SID |
| Method | | | | | | | | |
| GraN-DAG | **2.8 ± 2.5** | **7.5 ± 7.7** | 14.5 ± 5.2 | 52.6 ± 10.8 | 16.6 ± 5.3 | 103.6 ± 52.9 | 86.4 ± 21.6 | 1320.6 ± 145.8 |
| DAG-GNN | 10.1 ± 3.4 | 23.3 ± 11.5 | 18.3 ± 3.6 | 56.4 ± 6.1 | 45.5 ± 7.9 | 261.1 ± 88.8 | 224.3 ± 31.6 | 1741.0 ± 138.3 |
| NOTEARS | 11.1 ± 5.0 | 16.9 ± 11.3 | 20.3 ± 4.9 | 53.5 ± 10.5 | 53.7 ± 9.5 | 276.1 ± 96.8 | 201.8 ± 22.1 | 1813.6 ± 148.4 |
| CAM | **2.5 ± 2.0** | **7.9 ± 6.4** | **6.0 ± 5.6** | **29.3 ± 19.3** | **9.6 ± 5.1** | **39.0 ± 34.1** | **42.9 ± 6.6** | **857.0 ± 184.5** |
| GSF | 9.3 ± 3.9 | [13.9 ± 8.3 24.1 ± 12.5] | 29.5 ± 4.3 | [60.3 ± 11.6 75.0 ± 4.5] | 49.5 ± 5.1 | [151.5 ± 73.8 213.9 ± 82.5] | > 12 hours | |
| RANDOM | 23.0 ± 2.2 | 26.9 ± 18.1 | 33.5 ± 2.3 | 76.0 ± 6.2 | 689.7 ± 6.1 | 340.0 ± 113.6 | 711.5 ± 9.0 | 1916.2 ± 65.8 |

Table 14: Synthetic post nonlinear data sets

| | | PNL-GP | | PNL-MULT | |
|---|---|---|---|---|---|
| | | SHD | SID | SHD | SID |
| 10 nodes ER1 | GraN-DAG | **1.6±3.0** | **3.9±8.0** | **13.1±3.8** | 35.7±12.3 |
| | DAG-GNN | 11.5±6.8 | 32.4±19.3 | 17.900±6.2 | 40.700±14.743 |
| | NOTEARS | 10.7±5.5 | 34.4±19.1 | **14.0±4.0** | 38.6±11.9 |
| | CAM | **1.5±2.6** | **6.8±12.1** | 12.0±6.4 | 36.3±17.7 |
| | GSF | 6.2±3.3 | [7.7±8.7, 18.9±12.4] | 10.7±3.0 | [**9.8±11.9, 25.3±11.5**] |
| | RANDOM | 23.8±2.9 | 36.8±19.1 | 23.7±2.9 | 37.7±20.7 |
| 10 nodes ER4 | GraN-DAG | **10.9±6.8** | **39.8±21.1** | 32.1±4.5 | 77.7±5.9 |
| | DAG-GNN | 32.3±4.3 | 75.8±9.3 | 37.0±3.1 | 82.7±6.4 |
| | NOTEARS | 34.1±3.2 | 80.8±5.5 | 37.7±3.0 | 81.700±7.258 |
| | CAM | **8.4±4.8** | **30.5±20.0** | 34.4±3.9 | 79.6±3.8 |
| | GSF | 25.0±6.0 | [44.3±14.5, 66.1±10.1] | 31.3±5.4 | [**58.6±8.1, 76.4±9.9**] |
| | RANDOM | 35.0±3.3 | 80.0±5.1 | 33.6±3.5 | 76.2±7.3 |
| 50 nodes ER1 | GraN-DAG | 16.5±7.0 | 64.1±35.4 | **38.2±11.4** | 213.8±114.4 |
| | DAG-GNN | 56.5±11.1 | 334.3±80.3 | 83.9±23.8 | 507.7±253.4 |
| | NOTEARS | 50.1±9.9 | 319.1±76.9 | 78.5±21.5 | 425.7±197.0 |
| | CAM | **5.1±2.6** | **10.7±12.4** | **44.9±9.9** | 284.3±124.9 |
| | GSF | 31.2±6.0 | [59.5±34.1, 122.4±32.0] | 46.3±12.1 | [**65.8±62.2, 141.6±72.6**] |
| | RANDOM | 688.4±4.9 | 307.0±98.5 | 691.3±7.3 | 488.0±247.8 |
| 50 nodes ER4 | GraN-DAG | **68.7±17.0** | **1127.0±188.5** | **211.7±12.6** | 2047.7±77.7 |
| | DAG-GNN | 203.8±18.9 | 2173.1±87.7 | 246.7±16.1 | 2239.1±42.3 |
| | NOTEARS | 189.5±16.0 | 2134.2±125.6 | **220.0±9.9** | 2175.2±58.3 |
| | CAM | **48.2±10.3** | **899.5±195.6** | 208.1±14.8 | 2029.7±55.4 |
| | GSF | 105.2±15.5 | [1573.7±121.2, 1620±102.8] | > 12 hours | — |
| | RANDOM | 722.3±9.0 | 1897.4±83.7 | 710.2±9.5 | 1935.8±56.9 |

## A.7 METRICS

SHD takes two partially directed acyclic graphs (PDAG) and counts the number of edge for which the edge type differs in both PDAGs. There are four edge types: $i \leftarrow j$, $i \rightarrow j$, $i - j$ and $i \quad j$. Since this distance is defined over the space of PDAGs, we can use it to compare DAGs with DAGs, DAGs with CPDAGs and CPDAGs with CPDAGs. When comparing a DAG with a CPDAG, having $i \leftarrow j$ instead of $i - j$ counts as a mistake.

SHD-C is very similar to SHD. The only difference is that both DAGs are first mapped to their respective CPDAGs before measuring the SHD.

Introduced by Peters & Bühlmann (2015), SID counts the number of interventional distribution of the form $p(x_i | do(x_j = \hat{x}_j))$ that would be miscalculated using the *parent adjustment formula* (Pearl, 2009) if we were to use the predicted DAG instead of the ground truth DAG to form the parent adjustment set. Some care needs to be taken to evaluate the SID for methods outputting a CPDAG such as GES and PC. Peters & Bühlmann (2015) proposes to report the SID of the DAGs which have approximately the minimal and the maximal SID in the Markov equivalence class given by the CPDAG. See Peters & Bühlmann (2015) for more details.

## A.8    Hyperparameters

All GraN-DAG runs up to this point were performed using the following set of hyperparameters. We used RMSprop as optimizer with learning rate of $10^{-2}$ for the first subproblem and $10^{-4}$ for all subsequent suproblems. Each NN has two hidden layers with 10 units (except for the real and pseudo-real data experiments of Table 3 which uses only 1 hidden layer). Leaky-ReLU is used as activation functions. The NN are initialized using the initialization scheme proposed in Glorot & Bengio (2010) also known as *Xavier initialization*. We used minibatches of 64 samples. This hyperparameter combination have been selected via a small scale experiment in which many hyperparameter combinations have been tried manually on a single data set of type ER1 with 10 nodes until one yielding a satisfactory SHD was obtained. Of course in practice one cannot select hyperparameters in this way since we do not have access to the ground truth DAG. In Appendix A.9, we explain how one could use a held-out data set to select the hyperparameters of score-based approaches and report the results of such a procedure on almost settings presented in this paper.

For NOTEARS, DAG-GNN, and GSF, we used the default hyperparameters found in the authors code. It (rarely) happens that NOTEARS and DAG-GNN returns a cyclic graph. In those cases, we removed edges starting from the weaker ones to the strongest (according to their respective weighted adjacency matrices), stopping as soon as acyclicity is achieved (similarly to what was explained in Appendix A.2 for GraN-DAG). For GES and PC, we used default hyperparameters of the `pcalg` R package. For CAM, we used the the default hyperparameters found in the `CAM` R package, with default PNS and DAG pruning.

## A.9    Hyperparameter Selection via Held-out Score

Most structure/causal learning algorithms have hyperparameters which must be selected prior to learning. For instance, NOTEARS and GES have a regularizing term in their score controlling the sparsity level of the resulting graph while CAM has a thresholding level for its pruning phase (also controlling the sparsity of the DAG). GraN-DAG and DAG-GNN have many hyperparameters such as the learning rate and the architecture choice for the neural networks (i.e. number of hidden layers and hidden units per layer). One approach to selecting hyperparameters in practice consists in trying multiple hyperparameter combinations and keeping the one yielding the best score evaluated on a held-out set (Koller & Friedman, 2009, p. 960). By doing so, one can hopefully avoid finding a DAG which is too dense or too sparse since if the estimated graph contains many spurious edges, the score on the held-out data set should be penalized. In the section, we experiment with this approach on almost all settings and all methods covered in the present paper.

**Experiments:** We explored multiple hyperparameter combinations using random search (Bergstra & Bengio, 2012). Table 15 to Table 23 report results for each dataset types. Each table reports the SHD and SID averaged over 10 data sets and for each data set, we tried 50 hyperparameter combinations sampled randomly (see Table 24 for sampling schemes). The hyperparameter combination yielding the best held-out score among all 50 runs is selected *per data set* (i.e. the average of SHD and SID scores correspond to potentially different hyperparameter combinations on different data sets). 80% of the data was used for training and 20% was held out (GraN-DAG uses the same data for early stopping and hyperparameter selection). Note that the held-out score is always evaluated without the regularizing term (e.g. the held-out score of NOTEARS was evaluated without its L1 regularizer).

The symbols $^{++}$ and $^{+}$ indicate the hyperparameter search improved performance against default hyperparameter runs above one standard deviation and within one standard deviation, respectively. Analogously for $^{--}$ and $^{-}$ which indicate a performance reduction. The flag $_{***}$ indicate that, on average, less than 10 hyperparameter combinations among the 50 tried allowed the method to

converge in less than 12 hours. Analogously, $_{**}$ indicates between 10 and 25 runs converged and $_*$ indicates between 25 and 45 runs converged.

**Discussion:** GraN-DAG and DAG-GNN are the methods benefiting the most from the hyperparameter selection procedure (although rarely significantly). This might be explained by the fact that neural networks are in general very sensitive to the choice of hyperparameters. However, not all methods improved their performance and no method improves its performance in all settings. GES and GSF for instance, often have significantly worse results. This might be due to some degree of model misspecification which renders the held-out score a poor proxy for graph quality. Moreover, for some methods the gain from the hyperparameter tuning might be outweighed by the loss due to the 20% reduction in training samples.

**Additional implementation details for held-out score evaluation:** GraN-DAG makes use of a final pruning step to remove spurious edges. One could simply mask the inputs of the NN corresponding to removed edges and evaluate the held-out score. However, doing so yields an unrepresentative score since some masked inputs have an important role in the learned function and once these inputs are masked, the quality of the fit might greatly suffer. To avoid this, we retrained the whole model from scratch on the training set with the masking fixed to the one recovered after pruning. Then, we evaluate the held-out score with this retrained architecture. During this retraining phase, the estimated graph is fixed, only the conditional densities are relearned. Since NOTEARS and DAG-GNN are not always guaranteed to return a DAG (although they almost always do), some extra thresholding might be needed as mentioned in Appendix A.8. Similarly to GraN-DAG's pruning phase, this step can seriously reduce the quality of the fit. To avoid this, we also perform a retraining phase for NOTEARS and DAG-GNN. The model of CAM is also retrained after its pruning phase prior to evaluating its held-out score.

Table 15: Gauss-ANM - 10 nodes with hyperparameter search

| Graph Type | ER1 | | ER4 | | SF1 | | SF4 | |
|---|---|---|---|---|---|---|---|---|
| Metrics Method | SHD | SID | SHD | SID | SHD | SID | SHD | SID |
| GraN-DAG | $\mathbf{1.0 \pm 1.6}^+$ | $\mathbf{0.4 \pm 1.3}^{++}$ | $\mathbf{5.5 \pm 2.8}^+$ | $\mathbf{9.7 \pm 8.0}^{++}$ | $\mathbf{1.3 \pm 1.8}^-$ | $\mathbf{3.0 \pm 3.4}^+$ | $\mathbf{9.6 \pm 4.5}^+$ | $\mathbf{15.1 \pm 6.1}^+$ |
| DAG-GNN | $10.9 \pm 2.6^+$ | $35.5 \pm 13.6^+$ | $38.3 \pm 2.9^{--}$ | $84.4 \pm 3.5^-$ | $9.9 \pm 1.7^+$ | $30.3 \pm 18.8^-$ | $21.4 \pm 2.1^-$ | $44.0 \pm 15.5^+$ |
| NOTEARS | $26.7 \pm 6.9^{--}$ | $35.2 \pm 10.6^+$ | $20.9 \pm 6.6^{++}$ | $62.0 \pm 6.7^{++}$ | $20.4 \pm 9.6^{--}$ | $38.8 \pm 16.7^-$ | $26.9 \pm 7.4^-$ | $61.1 \pm 13.8^-$ |
| CAM | $3.0 \pm 4.2^-$ | $2.2 \pm 5.7^-$ | $\mathbf{7.7 \pm 3.1}^{++}$ | $23.2 \pm 14.7^+$ | $\mathbf{2.4 \pm 2.5}^-$ | $5.2 \pm 5.5^+$ | $\mathbf{9.6 \pm 3.1}^+$ | $20.1 \pm 6.8^-$ |
| GSF | $5.3 \pm 3.3^+$ | $[8.3 \pm 13.2^+$ $15.4 \pm 13.5]$ | $23.1 \pm 7.9^-$ | $[56.1 \pm 20.4^-$ $65.1 \pm 19.3]$ | $3.3 \pm 2.5^-$ | $[7.0 \pm 11.6^-$ $12.2 \pm 11.0]$ | $14.2 \pm 5.6^{--}$ | $[26.2 \pm 11.1^-$ $36.9 \pm 21.6]$ |
| GES | $38.6 \pm 2.1^{--}$ | $[20.3 \pm 15.4^+$ $28.3 \pm 18.4]$ | $33.0 \pm 3.4^-$ | $[66.2 \pm 7.0^+$ $76.6 \pm 4.3]$ | $38.3 \pm 2.4^{--}$ | $[8.8 \pm 5.2^-$ $25.5 \pm 18.2]$ | $33.6 \pm 4.8^{--}$ | $[32.7 \pm 12.7^-$ $52.0 \pm 14.0]$ |

Table 16: Gauss-ANM - 50 nodes with hyperparameter search

| Graph Type | ER1 | | ER4 | | SF1 | | SF4 | |
|---|---|---|---|---|---|---|---|---|
| Metrics Method | SHD | SID | SHD | SID | SHD | SID | SHD | SID |
| GraN-DAG | $\mathbf{3.8 \pm 3.3}^+$ | $\mathbf{15.0 \pm 14.0}^+$ | $105.6 \pm 16.5^-$ | $\mathbf{1131.7 \pm 91.0}^-$ | $24.7 \pm 6.4^+$ | $86.5 \pm 34.6^+$ | $112.7 \pm 15.5^{--}$ | $\mathbf{268.3 \pm 85.8}^+$ |
| DAG-GNN | $47.0 \pm 7.8^+$ | $268.1 \pm 118.0^+$ | $196.2 \pm 14.4^-$ | $1972.8 \pm 110.6^{++}$ | $51.8 \pm 5.6^-$ | $166.5 \pm 48.9^+$ | $144.2 \pm 11.6^+$ | $473.4 \pm 105.4^+$ |
| NOTEARS | $193.5 \pm 77.3^{--}$ | $326.0 \pm 99.1^+$ | $369.5 \pm 81.9^{--}$ | $2062.0 \pm 107.7^+$ | $104.8 \pm 22.4^{--}$ | $290.3 \pm 136.8^-$ | $213.0 \pm 35.1^{--}$ | $722.7 \pm 177.3^-$ |
| CAM | $\mathbf{4.0 \pm 2.7}^+$ | $\mathbf{21.1 \pm 22.1}^+$ | $105.6 \pm 20.9^-$ | $1225.9 \pm 205.7^-$ | $23.8 \pm 6.0^+$ | $\mathbf{81.5 \pm 15.3}^+$ | $112.2 \pm 14.0^-$ | $333.8 \pm 156.0^-$ |
| GSF | $24.9 \pm 7.4^*_*$ | $[40.0 \pm 26.3^-_*$ $77.5 \pm 45.3]$ | $129.3 \pm 20.4^-_*$ | $[1280.8 \pm 202.3^{--}_*$ $1364.1 \pm 186.7]$ | $35.3 \pm 6.9^-_*$ | $[99.7 \pm 41.7^-_*$ $151.9 \pm 59.7]$ | $121.6 \pm 11.7^-_{***}$ | $[310.8 \pm 108.1^-_{***}$ $391.9 \pm 93.3]$ |
| GES | $1150.1 \pm 9.8^{--}$ | $[112.7 \pm 71.1^+$ $132.0 \pm 89.0]$ | $1066.1 \pm 11.7^{--}$ | $[1394.3 \pm 81.8^{++}$ $1464.8 \pm 63.8]$ | $1161.7 \pm 7.0^{--}$ | $[322.8 \pm 211.1^-$ $336.0 \pm 215.4]$ | $1116.1 \pm 14.2^{--}$ | $[1002.7 \pm 310.9^{--}$ $1094.0 \pm 345.1]$ |

Table 17: Gauss-ANM - 20 nodes with hyperparameter search

| Graph Type | ER1 | | ER4 | | SF1 | | SF4 | |
|---|---|---|---|---|---|---|---|---|
| Metrics Method | SHD | SID | SHD | SID | SHD | SID | SHD | SID |
| GraN-DAG | $\mathbf{2.7 \pm 2.3}^+$ | $\mathbf{9.6 \pm 10.3}^+$ | $\mathbf{35.9 \pm 11.8}^+$ | $\mathbf{120.4 \pm 37.0}^{++}$ | $6.5 \pm 2.4^+$ | $\mathbf{17.5 \pm 6.3}^{++}$ | $35.6 \pm 4.1^+$ | $\mathbf{54.8 \pm 14.3}^+$ |
| DAG-GNN | $21.0 \pm 6.1^+$ | $98.8 \pm 42.2^+$ | $77.2 \pm 6.5^-$ | $345.6 \pm 18.6^-$ | $19.1 \pm 0.7^+$ | $55.0 \pm 20.1^+$ | $50.2 \pm 5.4^-$ | $118.7 \pm 33.2^-$ |
| NOTEARS | $101.5 \pm 39.6^{--}$ | $100.4 \pm 47.0^+$ | $124.0 \pm 16.3^{--}$ | $267.0 \pm 46.5^{++}$ | $55.0 \pm 28.2^{--}$ | $87.6 \pm 26.9^-$ | $66.7 \pm 8.3^{--}$ | $154.6 \pm 43.0^-$ |
| CAM | $\mathbf{2.8 \pm 2.2}^-$ | $\mathbf{11.5 \pm 10.2}^-$ | $64.3 \pm 29.3^-$ | $\mathbf{121.7 \pm 73.1}^+$ | $\mathbf{5.5 \pm 1.6}^-$ | $\mathbf{19.3 \pm 7.8}^+$ | $\mathbf{36.0 \pm 5.1}^-$ | $\mathbf{66.3 \pm 28.6}^-$ |
| GSF | $11.6 \pm 3.0^+$ | $[26.4 \pm 13.3^-$ $49.8 \pm 26.5]$ | $\mathbf{46.2 \pm 12.6}^-$ | $[172.7 \pm 40.8^-$ $213.5 \pm 38.6]$ | $12.8 \pm 2.1^{--}$ | $[32.1 \pm 14.0^{--}$ $56.2 \pm 13.8]$ | $42.3 \pm 5.1^-$ | $[68.9 \pm 27.7^-$ $95.1 \pm 33.8]$ |
| GES | $169.9 \pm 5.0^{--}$ | $[45.4 \pm 29.2^+$ $57.2 \pm 36.6]$ | $142.8 \pm 7.7^-_*$ | $[223.3 \pm 33.6^{++}_*$ $254.7 \pm 22.0]$ | $168.1 \pm 3.3^{--}$ | $[46.7 \pm 21.7^+$ $53.3 \pm 20.0]$ | $162.2 \pm 10.4^{--}$ | $[151.1 \pm 57.4^{--}$ $195.8 \pm 57.4]$ |

Table 18: Gauss-ANM - 100 nodes with hyperparameter search

| Graph Type | ER1 | | ER4 | | SF1 | | SF4 | |
|---|---|---|---|---|---|---|---|---|
| Metrics | SHD | SID | SHD | SID | SHD | SID | SHD | SID |
| Method | | | | | | | | |
| GraN-DAG | $\mathbf{15.1\pm7.5}^{+}$ | $\mathbf{65.1\pm33.2}^{+}$ | $\mathbf{191.6\pm17.8}^{+}$ | $4090.7\pm418.0^{+}$ | $51.6\pm10.2^{+}$ | $\mathbf{210.6\pm51.9}^{++}$ | $\mathbf{255.7\pm21.1}^{+}$ | $790.5\pm159.7^{+}$ |
| DAG-GNN | $103.9\pm9.1^{+}$ | $757.6\pm215.0^{+}$ | $387.1\pm25.3^{-}$ | $7741.9\pm522.5^{+}$ | $103.5\pm8.2^{-}$ | $391.7\pm60.0^{+}$ | $314.8\pm16.3^{+}$ | $1257.3\pm185.2^{+}$ |
| NOTEARS | $421.3\pm207.0^{--}$ | $945.7\pm339.7^{-}$ | $631.1\pm136.6^{--}$ | $8272.4\pm444.2^{-}$ | $244.3\pm63.8^{--}$ | $815.6\pm346.5^{-}$ | $482.3\pm114.1^{--}$ | $1929.7\pm363.1^{--}$ |
| CAM | $\mathbf{12.3\pm4.9}^{++}$ | $128.0\pm66.3^{-}$ | $\mathbf{198.8\pm22.2}^{-}$ | $4602.2\pm523.7^{-}$ | $51.1\pm9.4^{+}$ | $\mathbf{233.6\pm62.3}^{-}$ | $\mathbf{255.7\pm22.2}^{-}$ | $\mathbf{851.4\pm206.0}^{-}$ |
| GSF | $100.2\pm9.9_{**}^{--}$ | $[719.8\pm242.1_{**}^{--}$ $721.1\pm242.9]$ | $387.6\pm23.9_{***}$ | $[7535.1\pm595.2_{***}$ $7535.1\pm595.2]$ | $67.3\pm14.0_{***}^{+}$ | $[254.5\pm35.4_{***}^{--}$ $340.4\pm70.4]$ | $315.1\pm16.7_{***}^{-}$ | $[1214.0\pm156.4_{***}^{--}$ $1214.0\pm156.4]$ |
| GES | $4782.5\pm22.9^{--}$ | $[362.3\pm267.7^{+}$ $384.1\pm293.6]$ | $4570.1\pm27.9^{-}$ | $[5400.7\pm299.2^{++}$ $5511.5\pm308.5]$ | $4769.1\pm26.7^{--}$ | $[1311.1\pm616.6^{--}$ $1386.2\pm713.9]$ | $4691.3\pm47.3^{--}$ | $[3882.7\pm1010.6^{--}$ $3996.7\pm1075.7]$ |

Table 19: PNL-GP with hyperparameter search

| #Nodes | 10 | | | | 50 | | | |
|---|---|---|---|---|---|---|---|---|
| Graph Type | ER1 | | ER4 | | ER1 | | ER4 | |
| Metrics | SHD | SID | SHD | SID | SHD | SID | SHD | SID |
| Method | | | | | | | | |
| GraN-DAG | $\mathbf{1.2\pm2.2}^{+}$ | $\mathbf{1.9\pm4.2}^{+}$ | $\mathbf{9.8\pm4.9}^{+}$ | $\mathbf{29.0\pm17.6}^{+}$ | $12.8\pm4.9^{+}$ | $55.3\pm24.2^{+}$ | $73.9\pm16.8^{-}$ | $\mathbf{1107.2\pm144.7}^{+}$ |
| DAG-GNN | $10.6\pm4.9^{+}$ | $35.8\pm19.6^{-}$ | $38.6\pm2.0^{--}$ | $82.2\pm5.7^{--}$ | $48.1\pm8.4^{+}$ | $330.4\pm69.9^{+}$ | $192.5\pm19.2^{+}$ | $2079.5\pm120.9^{+}$ |
| NOTEARS | $20.6\pm11.4^{-}$ | $30.5\pm18.8^{+}$ | $24.2\pm6.5^{++}$ | $66.4\pm6.9^{++}$ | $102.1\pm27.3^{--}$ | $299.8\pm85.8^{+}$ | $660.0\pm258.2^{--}$ | $1744.0\pm232.9^{++}$ |
| CAM | $\mathbf{2.7\pm4.0}^{-}$ | $6.4\pm11.8^{+}$ | $\mathbf{8.7\pm4.5}^{-}$ | $30.9\pm20.4^{-}$ | $4.0\pm2.4^{+}$ | $10.7\pm12.4^{+}$ | $52.3\pm8.5^{-}$ | $913.9\pm209.3^{-}$ |
| GSF | $12.9\pm3.9_{***}^{--}$ | $[10.5\pm8.7_{***}^{---}$ $53.6\pm23.8]$ | $40.7\pm1.3_{**}^{-}$ | $[79.2\pm3.8_{**}^{-}$ $79.2\pm3.8]$ | $48.8\pm3.9_{**}^{--}$ | $[281.6\pm70.7_{**}^{--}$ $281.6\pm70.7]$ | $199.9\pm15.2_{***}$ | $[1878.0\pm122.4_{***}^{---}$ $1948.4\pm139.6]$ |

Table 20: PNL-MULT with hyperparameter search

| #Nodes | 10 | | | | 50 | | | |
|---|---|---|---|---|---|---|---|---|
| Graph Type | ER1 | | ER4 | | ER1 | | ER4 | |
| Metrics | SHD | SID | SHD | SID | SHD | SID | SHD | SID |
| Method | | | | | | | | |
| GraN-DAG | $10.0\pm4.5^{+}$ | $29.1\pm9.7^{+}$ | $32.9\pm3.3^{-}$ | $76.7\pm4.1^{+}$ | $\mathbf{59.8\pm28.2}^{-}$ | $213.6\pm97.3^{+}$ | $272.1\pm69.4^{-}$ | $2021.6\pm185.8^{+}$ |
| DAG-GNN | $14.6\pm3.1^{++}$ | $36.9\pm10.6^{+}$ | $38.9\pm2.0^{-}$ | $85.8\pm1.2^{--}$ | $\mathbf{64.3\pm27.8}^{+}$ | $508.8\pm317.2^{-}$ | $\mathbf{212.5\pm12.3}^{++}$ | $2216.9\pm95.6^{+}$ |
| NOTEARS | $28.8\pm9.1^{--}$ | $\mathbf{30.3\pm11.8}^{+}$ | $\mathbf{35.4\pm3.8}^{+}$ | $78.4\pm7.5^{+}$ | $160.2\pm67.5^{--}$ | $443.5\pm205.1^{-}$ | $229.2\pm25.4^{-}$ | $2158.8\pm70.3^{+}$ |
| CAM | $17.2\pm8.0^{-}$ | $33.7\pm14.4^{+}$ | $32.3\pm6.5^{+}$ | $76.6\pm8.2^{+}$ | $97.5\pm71.1^{-}$ | $\mathbf{282.3\pm123.8}^{+}$ | $251.0\pm25.9^{--}$ | $2026.2\pm58.2^{+}$ |
| GSF | $15.6\pm4.4_{**}^{--}$ | $[\mathbf{10.0\pm6.3}_{**}^{---}$ $\mathbf{60.1\pm17.2}]$ | $39.3\pm2.2_{**}^{-}$ | $[76.0\pm9.6_{**}^{--}$ $79.9\pm5.3]$ | $\mathbf{66.4\pm14.4}_{***}^{---}$ | $[145.1\pm96.1_{***}^{---}$ $618.8\pm257.0]$ | $> 12$ hours | |

Table 21: LIN with hyperparameter search

| #Nodes | 10 | | | | 50 | | | |
|---|---|---|---|---|---|---|---|---|
| Graph Type | ER1 | | ER4 | | ER1 | | ER4 | |
| Metrics | SHD | SID | SHD | SID | SHD | SID | SHD | SID |
| Method | | | | | | | | |
| GraN-DAG | $10.1\pm3.9^{-}$ | $\mathbf{28.7\pm14.7}^{-}$ | $34.7\pm2.9^{--}$ | $79.5\pm4.4^{-}$ | $40.8\pm10.3^{-}$ | $236.3\pm101.7^{+}$ | $256.9\pm55.7^{--}$ | $2151.4\pm144.3^{-}$ |
| DAG-GNN | $9.0\pm2.7^{++}$ | $35.6\pm11.4^{-}$ | $19.6\pm4.6^{+}$ | $63.9\pm7.5^{-}$ | $48.3\pm6.8^{+}$ | $381.7\pm145.4^{-}$ | $149.7\pm17.2^{++}$ | $2070.7\pm51.9^{--}$ |
| NOTEARS | $14.0\pm4.1_{--}^{-}$ | $32.2\pm7.9^{+}$ | $20.7\pm5.1^{++}$ | $63.1\pm8.0^{++}$ | $87.7\pm44.3^{-}$ | $294.3\pm99.3_{+}^{+}$ | $200.3\pm67.1^{--}$ | $\mathbf{1772.7\pm143.7}^{++}$ |
| CAM | $\mathbf{8.8\pm6.0}^{+}$ | $25.8\pm13.5^{+}$ | $33.9\pm2.8^{-}$ | $77.1\pm4.5^{+}$ | $\mathbf{34.8\pm7.0}^{+}$ | $221.2\pm98.3^{+}$ | $202.2\pm14.3^{--}$ | $1990.8\pm97.5^{-}$ |
| GSF | $10.7\pm3.5^{-}$ | $[15.8\pm8.4^{-}$ $45.2\pm20.2]$ | $33.4\pm3.3^{++}$ | $[71.7\pm11.5^{+}$ $77.3\pm6.1]$ | $54.4\pm6.5_{*}^{-}$ | $[158.1\pm115.9_{*}^{-}$ $560.9\pm220.7]$ | $195.6\pm9.9_{**}$ | $[2004.9\pm85.2_{**}^{-}$ $2004.9\pm85.2]$ |

Table 22: ADD-FUNC with hyperparameter search

| #Nodes | 10 | | | | 50 | | | |
|---|---|---|---|---|---|---|---|---|
| Graph Type | ER1 | | ER4 | | ER1 | | ER4 | |
| Metrics | SHD | SID | SHD | SID | SHD | SID | SHD | SID |
| Method | | | | | | | | |
| GraN-DAG | $\mathbf{2.6\pm2.4}^{+}$ | $\mathbf{4.3\pm4.3}^{+}$ | $7.0\pm3.1^{++}$ | $37.1\pm12.4^{++}$ | $13.2\pm6.7^{+}$ | $\mathbf{72.1\pm55.2}^{+}$ | $90.1\pm25.6^{-}$ | $1241.7\pm289.8^{+}$ |
| DAG-GNN | $8.7\pm2.8^{++}$ | $22.3\pm9.4^{+}$ | $25.3\pm3.8^{++}$ | $63.6\pm8.6^{++}$ | $44.7\pm9.7^{++}$ | $306.9\pm114.7^{+}$ | $194.0\pm20.4^{+}$ | $1949.3\pm107.1^{+}$ |
| NOTEARS | $21.2\pm11.5_{-}^{-}$ | $15.5\pm9.9_{*}^{+}$ | $13.3\pm4.3^{++}$ | $41.3\pm11.5^{++}$ | $186.8\pm83.0^{--}$ | $276.9\pm92.1^{-}$ | $718.4\pm170.4^{--}$ | $1105.9\pm250.1^{++}$ |
| CAM | $\mathbf{3.0\pm2.2}^{-}$ | $8.1\pm6.3^{-}$ | $6.2\pm5.5^{-}$ | $28.5\pm21.5^{+}$ | $10.0\pm4.6^{-}$ | $44.2\pm32.1^{-}$ | $46.6\pm9.5^{-}$ | $882.5\pm186.5^{-}$ |
| GSF | $5.5\pm4.1^{+}$ | $[7.5\pm12.3^{+}$ $16.3\pm12.9]$ | $19.1\pm7.0^{++}$ | $[44.5\pm19.7^{+}$ $60.4\pm16.5]$ | $29.8\pm7.6_{*}^{+}$ | $[\mathbf{44.6\pm42.6}_{*}^{++}$ $\mathbf{96.8\pm46.7}]$ | $140.4\pm31.7_{***}$ | $[1674.4\pm133.9_{***}$ $1727.6\pm145.2]$ |

Table 23: Results for real and pseudo real data sets with hyperparameter search

| Data Type | Protein signaling data set | | | SynTReN - 20 nodes | | |
| Metrics | SHD | SHD-C | SID | SHD | SHD-C | SID |
| Method | | | | | | |
|---|---|---|---|---|---|---|
| GraN-DAG | $12.0^+$ | $\mathbf{9.0}^+$ | $48.0^-$ | $41.2 \pm 9.6^-$ | $43.7 \pm 8.3^-$ | $144.3 \pm 61.3^+$ |
| GraN-DAG++ | $14.0^-$ | $11.0^-$ | $57.0^-$ | $46.9 \pm 14.9^-$ | $49.5 \pm 14.7^-$ | $158.4 \pm 61.5^-$ |
| DAG-GNN | $16.0$ | $14.0^+$ | $59.0^-$ | $\mathbf{32.2 \pm 5.0}^{++}$ | $\mathbf{32.3 \pm 5.6}^{++}$ | $194.2 \pm 50.2^-$ |
| NOTEARS | $15.0^+$ | $14.0^+$ | $58.0^-$ | $44.2 \pm 27.5^{++}$ | $45.8 \pm 27.7^{++}$ | $183.1 \pm 48.4^{--}$ |
| CAM | $\mathbf{11.0}^+$ | $9.0$ | $51.0^+$ | $101.7 \pm 37.2^{--}$ | $105.6 \pm 36.6^{--}$ | $\mathbf{111.5 \pm 25.3}^{++}$ |
| GSF | $20.0^-$ | $14.0^-$ | $[37.0^+$ $60.0]$ | $\mathbf{27.8 \pm 5.4}^{++}_{*}$ | $\mathbf{27.8 \pm 5.4}^{++}_{*}$ | $[207.6 \pm 55.4^{--}_{*}$ $209.6 \pm 59.1]$ |
| GES | $47.0^-$ | $50.0^-$ | $[\mathbf{37.0}^+$ $\mathbf{47.0}]$ | $167.5 \pm 5.6^{--}$ | $172.2 \pm 7.0^{--}$ | $[\mathbf{75.3 \pm 24.4}^{++}$ $\mathbf{97.6 \pm 30.8}]$ |

Table 24: Hyperparameter search spaces for each algorithm

| | Hyperparameter space |
|---|---|
| GraN-DAG | Log(learning rate) $\sim U[-2, -3]$ (first subproblem) 
 Log(learning rate) $\sim U[-3, -4]$ (other subproblems) 
 $\epsilon \sim U\{10^{-3}, 10^{-4}, 10^{-5}\}$ 
 Log(pruning cutoff) $\sim U\{-5, -4, -3, -2, -1\}$ 
 # hidden units $\sim U\{4, 8, 16, 32\}$ 
 # hidden layers $\sim U\{1, 2, 3\}$ 
 Constraint convergence tolerance $\sim U\{10^{-6}, 10^{-8}, 10^{-10}\}$ 
 PNS threshold $\sim U[0.5, 0.75, 1, 2]$ |
| DAG-GNN | Log(learning rate) $\sim U[-4, -2]$ 
 # hidden units in encoder $\sim U\{16, 32, 64, 128, 256\}$ 
 # hidden units in decoder $\sim U\{16, 32, 64, 128, 256\}$ 
 Bottleneck dimension (dimension of $Z$) $\sim U\{1, 5, 10, 50, 100\}$ 
 Constraint convergence tolerance $\sim U\{10^{-6}, 10^{-8}, 10^{-10}\}$ |
| NOTEARS | L1 regularizer coefficient $\sim U\{0.001, 0.005, 0.01, 0.05, 0.1, 0.5\}$ 
 Final threshold $\sim U\{0.001, 0.003, 0.01, 0.03, 0.1, 0.3, 1\}$ 
 Constraint convergence tolerance $\sim U\{10^{-6}, 10^{-8}, 10^{-10}\}$ |
| CAM | Log(Pruning cutoff) $\sim U[-6, 0]$ |
| GSF | Log(RKHS regression regularizer) $\sim U[-4, 4]$ |
| GES | Log(Regularizer coefficient) $\sim U[-4, 4]$ |

