# OpenReview forum: "Gradient-Based Neural DAG Learning"
_ICLR.cc/2020/Conference — Accept (Poster)_

### Official Review · AnonReviewer1 · 2019-10-22
**Official Blind Review #1**

**Rating:** 8

**Review:**

################## Response to rebuttal ##################

I would like to thank the authors for their diligent rebuttal. Overall, I consider that the issues raised in my review have been mostly addressed.

In summary, after reading the other reviews and going through the revised manuscript, I believe the paper would be a meaningful contribution to the field, both in terms of the potential applicability of GraN-DAG to certain problems and its potential to lead to future work. As a result, I have updated my score to support acceptance without reservations.

In particular, the authors now discuss clearly some of the potential limitations of the approach, such as the scaling of its asymptotic complexity with respect to the number of features or its statistical performance in the small sample size regime. The additional experimental results are, in my opinion, also a good addition to the manuscript, further clarifying the strengths and weaknesses of NOTEARS, CAM and GraN-DAG under different regimes and modelling assumptions. I believe these changes substantially improve the paper, as it will both help practitioners decide when to use GraN-DAG as well as clearly delineate avenues for future work.

# Minor points / additional questions

The new results regarding convergence in Table 4 suggest that early stopping might indeed be instrumental. I wonder the extent to which NOTEARS could benefit from early stopping, be it in terms of runtime or, less likely due to its use of explicit regularization, the quality of the structures it retrieves.

# Typos

Section 3.6: … GraN-DAG performs fewer iteration(s) than NOTEARS…
Section 5: GraN-DAGand DAG-GNN -was- were not designed with…
Appendix A.1: … graphs can be roughly -halfed- halved when executed…
Appendix A.5: … graphs were generated using the -BarabsiAlbert- Barabási-Albert model …
Appendix A.9: … to selecting hyperparameters in practice consist(s) in ...

################## Original review ##################

# High-level assessment

The manuscript presents an alternative to the method in [2] to extend the ideas of [1] to nonlinear DGMs using neural networks.

From a methodological perspective, in my opinion the proposed approach is sound and sufficiently novel to be a significant contribution. While some limitations might be present, such as being prone to overfitting without relying on additional (pre/post)-processing heuristics or perhaps being computationally intensive for high-dimensional data ($O(p^{3})$ runtime per SGD step), I believe that addressing those issues (if indeed present) could be left for future work.

The paper is written clearly, being easy-to-follow while providing enough detail to understand the proposed method in depth and attempt to reproduce the results.

In my opinion, perhaps the weakest aspect of the paper concerns the experimental analysis, which is somewhat limited in its scope. Most notably, I believe the current experimental setup focuses on cases likely to be favorable to the proposed approach relative to the baselines ($n > p$, nonlinear and non-additive ground-truth models) while presenting no results for other settings commonly encountered in structure learning applications, such as high-dimensional $n << p$ problems or ground-truth models given by linear or generalized linear SCMs.

Taking all those factors into consideration, I currently lean towards a “weak accept” rating, but would gladly modify my score after the author discussion phase as appropriate.

# Summary

In this paper, the authors propose a novel approach for learning the structure of Directed Graphical Models (DGMs) from observational data that allows to flexibly model nonlinear relationships between variables using neural networks.

The proposed method builds on the recent breakthrough in [1], which pioneered the use of a smooth characterization of acyclicity for directed graphs. In particular, the authors in [1] use this characterization to learn the structure of **linear** DGMs by solving a (nonconvex) continuous, constrained optimization problem.

The main contribution of this manuscript is an extension of the approach in [1] to the nonlinear setting by making use of neural networks, and which appears to outperform the only existing method that shares a similar high-level goal [2], presumably by allowing more flexibility in its parametrization.

In a nutshell, the authors accomplish this by:

1. Flexibly modelling the conditional distribution of each variable $X_{j}$ as a nonlinear function of (possibly) all other variables $\bf{X_{-j}}$ using an MLP, i.e. $p_{j}(X_{j} \mid \bf{X_{-j}}) = f_{j}(X_{j} ; NN_{j}(\bf{X_{-j}}))$, with $f_{j}$ denoting any probability density differentiable w.r.t. its parameters and $NN_{j}$ being a neural network with parameters $\theta_{j}$.

2. Deriving a function $C_{i,j}(\theta_{j})$ such that $C_{i,j}(\theta_{j}) = 0$ is a sufficient condition for the conditional distribution of $X_{j}$ to be independent of $X_{i}$.

3. Using $C_{i,j}(\theta_{j})$, which is differentiable w.r.t. $\theta_{j}$ for most values of $\theta_{j}$, to define a “connectivity” matrix to be directly plugged into the acyclicity criterion in [1].

4. Following [1], the model is then fit by (constrained) maximum likelihood using an augmented Lagrangian method. However, unlike [1], the unconstrained subproblem in each iteration of the augmented Lagrangian method is solved using stochastic gradient descent, rather than a batch quasi-Newton method, and regularized using early stopping on an external validation set, rather than by using explicit $L_{1}$ regularization to induce sparsity.

5. The authors also incorporate additional “heuristics”, such as borrowing the Preliminary Neighbour Selection (PNS) and feature selection pruning techniques from [3], which are reported to greatly enhance the performance of the proposed approach (Table 4), and adapt the thresholding scheme in [1] to the nonlinear setting.

The proposed method is evaluated using synthetic and semi-synthetic data, as well as one real-world dataset.  The results suggest that, in low dimensional problems ($n > p$, $p \le 100$) for which the ground-truth relationships between variables are nonlinear and non-additive, the proposed approach clearly outperforms [2] and also outperforms [3] in many settings, albeit often not by a statistically significant margin in the latter case.

# Major points / suggestions

1. I believe it could be argued that one of the main drawbacks of the smooth acyclicity constraint in [1] is its high computational complexity w.r.t. the number of features. However, [1] uses a batch quasi-Newton method to solve the unconstrained subproblems in their optimization routine, which should in principle lead to relatively few $O(p^3)$ matrix exponential evaluations being necessary.

As it relies on the same characterization of acyclicity as [1], this drawback might also limit the scalability of the proposed approach for high-dimensional data. Moreover, since the method in this manuscript uses mini-batch gradient descent instead, this problem could in fact be exacerbated by perhaps requiring substantially more $O(p^3)$ matrix exponential evaluations.

To this end, I would be glad if the authors could clarify whether this could indeed be a limitation or not. In any case, I believe it could also be helpful to include a small discussion regarding computational complexity for the proposed approach and related work in the manuscript or appendix.

2. All experimental results reported in the manuscript use only data generation models with nonlinear and non-additive relationships between variables.

This setup is well aligned with the contribution, as it emphasises its ability to flexibly model nonlinear relationships between variables learnt using global, gradient-based optimization. In contrast, most baselines are limited to either linear/generalized linear models or use some variant of greedy search.

While investigating this setting is certainly the main priority given the problem statement, I believe it would also be important to characterize the extent to which the performance of the proposed approach deteriorates when this extra flexibility is not needed, if at all. To this end, I would encourage the authors to repeat their experiments for synthetic data, but generated under linear and generalized linear SCMs.

In particular, it is remarkable that CAM [3], which suffers from model misspecification due assuming the data is generated by a generalized linear model, remains so competitive throughout all experiments shown in the current version of the manuscript. It would be interesting to see if the proposed approach can match or still outperform CAM under CAM’s “ideal scenario”.

3. Parallel to the previous point, all experimental results in the manuscript so far concern somewhat low-dimensional data $p \le 100$ and relatively abundant data $n \approx 1,000 > p$.

This setup might also be well aligned with the proposed approach, which could be more prone to overfitting than some of the baselines due to its high capacity and its lack of explicit sparsity-inducing regularization, which was found to boost performance for NOTEARS [1, Fig. 3(b)].

Also, in the small sample size regime, the reliance on an external validation set for early stopping might become a liability by (i) reducing the amount of data available for model fitting and (ii) making the fitted DGMs less stable w.r.t. changes in the random seed which, compounded with the nonconvexity of the objective, could be problematic if the DGM is to be interpreted for downstream tasks (e.g. in computational biology).

To this end, I would likewise encourage the authors to extend their experimental setup for synthetic data by (i) repeating their experiments for one “high-dimensional” setting ($n << p$) and (ii) studying the stability of the DAG w.r.t. different training runs.

4. All experimental results currently focus heavily on accuracy, while the computational aspect of the problem is largely secondary.

To this end, it would be interesting to study, using synthetic data, the scalability of the proposed approach relative to the best performing baseline approaches with respect to (i) number of features, (ii) sparsity of the ground-truth DGM and (iii) sample size, covering a sufficiently broad range in each case (low-dimensional vs high-dimensional, sparse vs dense, scarce vs abundant sample size).

# Minor points / suggestions

1. Perhaps the decision to permanently set the thresholding masks to 0 when an entry of the connectivity matrix is “small enough” could be justified further. For example, by showing that this heuristic rarely “zeros out” incorrectly any entries, or that such entries would seldom become nonzero again if the model was trained for longer without being masked.

2. I might have missed it, but I could not find the info about how the train/validation split is formed for the experiments following Section A.1.

# References

[1] Zheng, Xun, et al. "DAGs with no tears: Continuous optimization for structure learning." *Advances in Neural Information Processing Systems*. 2018.
[2] Yu, Yue, et al. "DAG-GNN: DAG Structure Learning with Graph Neural Networks." *International Conference on Machine Learning*. 2019.
[3] Bühlmann, Peter, Jonas Peters, and Jan Ernest. "CAM: Causal additive models, high-dimensional order search and penalized regression." *The Annals of Statistics* 42.6 (2014): 2526-2556.

**Experience Assessment:**

I have read many papers in this area.

**Review Assessment: Checking Correctness Of Derivations And Theory:**

I carefully checked the derivations and theory.

**Review Assessment: Checking Correctness Of Experiments:**

I carefully checked the experiments.

**Review Assessment: Thoroughness In Paper Reading:**

I read the paper thoroughly.

---

> ### Author Response · Authors · 2019-11-12
> **Answer to reviewer #1**
>
> We thank reviewer 1 for their very thorough review. Many good points have been raised and we believe we managed to answer the most important ones.
>
> Our updated paper contains many new experiments many of which appears in the Appendix. With the added clarifications, the paper now has a little more than 9 pages. The present version is a proposition. We are open to relocate certain sections or tables if reviewers believe it would improve the paper.
>
> *Computational complexity*
>
> We have added a brief discussion on the computational complexity of GraN-DAG (Section 3.6 of the main paper). We are currently working to provide some running times.
>
> *Experiments on linear and additive synthetic data sets*
>
> We have added experiments on linear Gaussian data and on data satisfying CAM assumptions (function additivity) in Appendix A.6 (discussion in Section 4.1 of main paper). On linear data, GraN-DAG and CAM performs similarly to NOTEARS. On additive function data, CAM dominates all methods and GraN-DAG is the second best algorithm.
>
> *High-dimensional setting*
>
> We do not expect GraN-DAG to yield good results in the high-dimensional setting (p >> n) since the problem of fitting a neural network with less data than features has not been explored very much. We expect severe overfitting due to the lack of data and, as a consequence, a poor DAG recovery. We recognize the paper was not mentioning this fact and decided to include a small discussion in our updated version in Section 5 of the main paper, in which we clarify the scope of problems to which GraN-DAG can be applied.
>
> *CAM is on par/better than GraN-DAG on nonlinear Gaussian additive noise model. Why?*
>
> In Section 4.1, we propose an explanation for this fact and refer to Appendix A.4 for supporting evidence. We believe that, with sample sizes considered in this paper (which are typical in the structure learning litterature), GraN-DAG has a higher variance than CAM due to its improved capacity, thus partly offsetting the benefits of its greater flexibility. Our small experiment seems to suggest GraN-DAG can outperform CAM in larger sample size settings, which supports our hypothesis.

---

> > ### Author Response · Authors · 2019-11-15
> > **Computational complexity**
> >
> > *We have nuanced our computational complexity discussion in Section 3.6*
> >
> > New data (Appendix A.1 Table 4) shows GraN-DAG takes fewer iterations than NOTEARS in total before the augmented Lagrangian converges, even though GraN-DAG uses a stochastic gradient method. We hypothesize this is due to early stopping which avoids having to wait until complete convergence before moving to the next subproblem. GraN-DAG runtime is nevertheless greater than NOTEARS due to its more involved model architecture (total runtime $\approx$ 10 minutes against $\approx$ 1 minute for 20 nodes graphs and $\approx$ 4 hours against $\approx$ 1 hour for 100 nodes graphs). CAM requires $\approx$ 20 minutes for both 20 and 100 nodes graphs. GraN-DAG runtime can be reduced roughly by a half if executed on GPU.

---

### Official Review · AnonReviewer2 · 2019-10-23
**Official Blind Review #2**

**Rating:** 6

**Review:**

This work addresses the problem of learning the structure of directed acyclic graphs in the presence of nonlinearities. The proposed approach is an extension of the NOTEARS algorithm which uses a neural network for each node in the graph during structure learning. This adaptation allows for non-linear relationships to be easily modeled. In addition to the proposed adaptation, the authors employ a number of heuristics from the causal discovery literature to improve the efficacy of the search. Empirical results are provided which compare the proposed algorithm to prior art.

Overall, I found the paper to be well written and sensible. However, there are a few items that prevent me from recommending acceptance at this time:

1) The proposed approach seems to necessitate a fair number of hyperparameters. How were these chosen? On what basis should practitioners choose their hyperparameters for real-world application?

2) The authors employ causal language in both the introduction and the experiment section. However, there is no notion of (a) whether the proposed algorithm is sound and complete,  and (b) under what assumptions we can expect a fully directed graph to be reasonable (I assume the additive noise model?).

3) The experiments seem to indicate that prior work (CAM) outperforms or performs similarly to the proposed method. While this shouldn't prevent a paper from being published, it would be nice to see an extended discussion as to why the authors think this is occuring.

**Experience Assessment:**

I have published one or two papers in this area.

**Review Assessment: Checking Correctness Of Derivations And Theory:**

I carefully checked the derivations and theory.

**Review Assessment: Checking Correctness Of Experiments:**

I assessed the sensibility of the experiments.

**Review Assessment: Thoroughness In Paper Reading:**

I read the paper thoroughly.

---

> ### Author Response · Authors · 2019-11-12
> **Answer to reviewer #2**
>
> We thank reviewer 2 for their review. We believe these comments allowed us to greatly improve the paper.
>
> Our updated paper contains many new experiments many of which appears in the Appendix. With the added clarifications, the paper now has a little more than 9 pages. The present version is a proposition. We are open to relocate certain sections or tables if reviewers believe it is necessary.
>
> *Hyperparameter selection*
>
> The hyperparameters of GraN-DAG have been selected via a small scale experiment in which many hyperparameter combinations have been tried manually on a single data set of type ER1 with ten nodes. The hyperparameters yielding a satisfactory SHD were retained and used for all other experiments (except the real and pseudo real experiments which used one hidden layer instead of two). All other methods used default hyperparameters. We made this more explicit in the paper (Appendix A.8).
>
> One approach to selecting hyperparameters in practice consist in trying multiple hyperparameter combinations and keeping the one yielding the best score evaluated on a held-out set (Koller & Friedman, 2009, p.  960). We have performed such a procedure for all data sets and all score-based approaches and are reporting the results in Table 14 to 22 in the appendix of our updated paper. The details of the hyperparameter search can be found in Appendix A.9. The procedure does not seem to alter our conclusions significantly.
>
> *Theory*
>
> In Section 3.3, we provide more details regarding the assumptions under which GraN-DAG would recover the correct graph. We contrast our analysis with the guarantees of CAM and NOTEARS in Section 5. In a nutshell, if we could solve exactly problem (10) in its population form (which we can not, due to non-convexity), we would recover the correct graph when the graph is identifiable from the distribution. The synthetic data of Section 4.1 with nonlinear functions and Gaussian noise is an example of an identifiable setup.
>
> *CAM is on par/better than GraN-DAG on nonlinear Gaussian additive noise model. Why?*
>
> In Section 4.1, we propose an explanation for this fact and refer to Appendix A.4 for supporting evidence. We believe that, with sample sizes considered in this paper (which are typical in the structure learning litterature), GraN-DAG has a higher variance than CAM due to its improved capacity, thus partly offsetting the benefits of its greater flexibility. Our small experiment seems to suggest GraN-DAG can outperform CAM in larger sample size settings, which supports our hypothesis.

---

### Official Review · AnonReviewer3 · 2019-10-26
**Official Blind Review #3**

**Rating:** 6

**Review:**

Summary:
The authors propose a prediction model for directed acyclic graphs (DAGs) over a fixed set of vertices based on a neural network. The present work follows the previous work on undirected acyclic graphs, where the key constraint is (3), ensuring the acyclic property. The proposed method performed favorably on artificial/real data compared to previous baselines.

Comments:
I do not understand yet if the proposed formulation really ensures the acyclic condition. More precisely, the condition (3) ensures undirected acyclic property, which also implies the directed ones. However, I am afraid that the condition (3) might also eliminate DAGs containing “undirected” cycles, i.e., cycles when we neglect directions of edges as well. So, I think a formal proof is necessary to show that the proposed formulation can output all possible DAGs, not a subset.

The present work heavily relies on the previous work of Zheng et al. (2018). The proposed formulation is designed for DAGs, but due to my concern about the correctness raised above, it is difficult for me to evaluate the originality of the present work.

Minor comments:
-“X_\pi_j^G denote the random vector containing the variables corresponding to the parents of j in G.” – this definition is not clear. Please clarify the meaning.


Comments after Rebuttal:
After reading the rebutall comments, I modified my score.

**Experience Assessment:**

I have published in this field for several years.

**Review Assessment: Checking Correctness Of Derivations And Theory:**

I assessed the sensibility of the derivations and theory.

**Review Assessment: Checking Correctness Of Experiments:**

I assessed the sensibility of the experiments.

**Review Assessment: Thoroughness In Paper Reading:**

I read the paper at least twice and used my best judgement in assessing the paper.

---

> ### Author Response · Authors · 2019-11-08
> **Answer to reviewer #3**
>
> We thank reviewer #3 for their review of our paper. We hope the reviewer can update their review which is currently based on an important misunderstanding corrected below.
>
> *TL;DR*
>
> Important misunderstanding to clarify: the condition (3) ensures directed acyclicity -- only directed cycles are penalized (and note that Zheng et al. (2018) were always only talking about directed cycles). We provide more explanations below to clarify this point.
>
> *About Zheng et al. (2018)*
>
> Zheng et al. (2018) proposes a score-based algorithm to learn directed acyclic graphs from observations using continuous constrained optimization. To do so, the graph is encoded as a weighted adjacency matrix $U = [u_1|u_2| ...|u_d] \in \mathbb{R}^{d \times d}$ which represents coefficients in a linear SEM of the form $X_j := u_j^\intercal X + N_j\ \ \forall j \in V$ where $N_j$ is a noise variable. Note that this induces a directed graph $\mathcal{G}_U$ in which the edge $i \rightarrow j$ is present iff $U_{ij} \not= 0$. For an arbitrary $U$, the directed graph $\mathcal{G}_U$ will not necessarily be acyclic. For instance, if $U_{ii} \not= 0$ we have a directed cycle of length one from node $i$ to itself. To enforce acyclicity, the authors propose the constraint
> $$\text{Tr}\exp U \odot U - d = 0$$
> Our paper provides a partial explanation for why this constraint is in fact characterizing directed acyclicity and refers to Zheng et al. (2018) for a thorough explanation.
>
> *Constraint avoids only directed cycles*
>
> Reviewer #3 fears this constraint is "too strong" in the sense that, additionally to avoiding directed cycles, the constraint also avoids undirected cycles. We would like to stress the fact that this concern is about the work of Zheng et al. (2018) and not precisely our contribution. Given the importance of this result in our own work, we, nevertheless, provide an explanation for why the constraint is in fact avoiding only directed cycles while allowing undirected cycles.
>
> Consider a setup with 3 variables. Let $B := U \odot U$. Using the definition of the matrix multiplication, we have that
>
> $$(B^3)_{ii} = \sum_{k\ t} B_{ik}B_{kt}B_{ti}\ \ \forall i$$
> where $B_{ik}B_{kt}B_{ti} \geq 0\ \ \forall i,k,t$. Since each element of $B$ is non-negative and using the definition of the trace and the matrix exponential we have that
> $$\frac{(B^3)_{ii}}{3!} \leq \text{Tr}\exp B - d \ \ \forall i$$
> Note that each term in this sum corresponds to a potential directed cycle from $i$ to $i$ of length 3. If for instance the directed cycle $i \rightarrow k_0 \rightarrow t_0 \rightarrow i$ is present, then we have that $B_{ik_0}B_{k_0t_0}B_{t_0i} > 0$ which implies that $\text{Tr}\exp B - d > 0$ i.e. the constraint will not be satisfied. However, if the undirected cycle $i \rightarrow k_0 \leftarrow t_0 \rightarrow i$ is present, then it is true that $B_{ik_0}B_{t_0k_0}B_{t_0i} > 0$, but this term does not appear in the sum, hence it does not have any effect on the constraint. In fact no "undirected cycle" terms appear in the expression of $\text{Tr}\exp B - d$, meaning the constraint does not concern undirected cycles. For a formal proof, we refer reviewer 3 to Zheng et al. (2018).

---

### Author Response · Authors · 2019-11-15
**Final submission update**

This message is to inform the reviewers that our paper submission has been updated a second time. In this version, we have sharpened our motivation and how it relates to causality, clarified the connection between our experiments and theoretical guarantees, nuanced our computational complexity discussion and improved the hyperparameter selection section in appendix. Other modifications are minor clarifications.

The paper now makes 9 pages and a half. We would like to remind the reviewers that we are open to relocate certain sections or tables if they think it would improve the present submission.

Again, we thank the reviewers for the work they have put into reviewing this paper.

---

### Decision · Program_Chairs · 2019-12-19

**Decision:**

Accept (Poster)

**Comment:**

In this paper, the authors propose a novel approach for learning the structure of a directed acyclic graph from observational data that allows to flexibly model nonlinear relationships between variables using neural networks. While the reviewers initially had concerns with respect to the positioning of the paper and various questions regarding theoretical results and experiments, these concerns have been addressed satisfactorily during the discussion period.  The paper is now acceptable for publication in ICLR-2020.